# Spectrum Random Masking for Generalization in Image-based Reinforcement Learning

**Yangru Huang**[1], **Peixi Peng**[1] [2] *, **Yifan Zhao**[1], **Guangyao Chen**[1], **Yonghong Tian**[1] [2] *
[1]School of Computer Science, Peking University, Beijing, China
[2]Peng Cheng Laboratory, ShenZhen, China
yrhuang@stu.pku.edu.cn, {pxpeng, zhaoyf, gychen, yhtian}@pku.edu.cn

## Abstract

Generalization in image-based reinforcement learning (RL) aims to learn a robust policy that could be applied directly on unseen visual environments, which is a challenging task since agents usually tend to overfit to their training environment. To handle this problem, a natural approach is to increase the data diversity by image based augmentations. However, different with most vision tasks such as classification and detection, RL tasks are not always invariant to spatial based augmentations due to the entanglement of environment dynamics and visual appearance. In this paper, we argue with two principles for augmentations in RL: *First*, the augmented observations should facilitate learning a universal policy, which is robust to various distribution shifts. *Second*, the augmented data should be invariant to the learning signals such as action and reward. Following these rules, we revisit image-based RL tasks from the view of frequency domain and propose a novel augmentation method, namely Spectrum Random Masking (SRM),which is able to help agents to learn the whole frequency spectrum of observation for coping with various distributions and compatible with the pre-collected action and reward corresponding to original observation. Extensive experiments conducted on DMControl Generalization Benchmark demonstrate the proposed SRM achieves the state-of-the-art performance with strong generalization potentials.

## 1 Introduction

Reinforcement Learning (RL) from image-based observations is a task where representation and decision-making are jointly learned from visual signals. Although this type of RL methods has made significant progresses due to the development of deep Convolution Neural Networks (CNNs), most of them suffer from drastic performance decline on previously unseen environments [24, 44, 4], especially for image-based inputs [44, 32, 4, 48, 27]. The reason is that classical RL methods assume the environments of training and testing are identical [17]. However, there usually exists significant distributions divergences between them in real-world scenarios. The high dimensionality of visual inputs coupled with the shifted distributions of environment make the generalization of RL tasks even more challenging [30].

To handle this challenge, one natural way is to decrease the distribution gap of training and testing data by simulating the diversity of test environments in training. Recent studies show that the appropriate data augmentation could facilitate the generalization of the policy network [41]. Most of these data augmentation methods have been carried out directly in the spatial domain [19], such as flipping and rotation. However, the dynamic of environment has the disadvantages of lacking sensitivity to such spatial-based data augmentation [28]. An example is illustrated in Fig. 1: when horizontal flipping is

---

*Corresponding author

36th Conference on Neural Information Processing Systems (NeurIPS 2022).

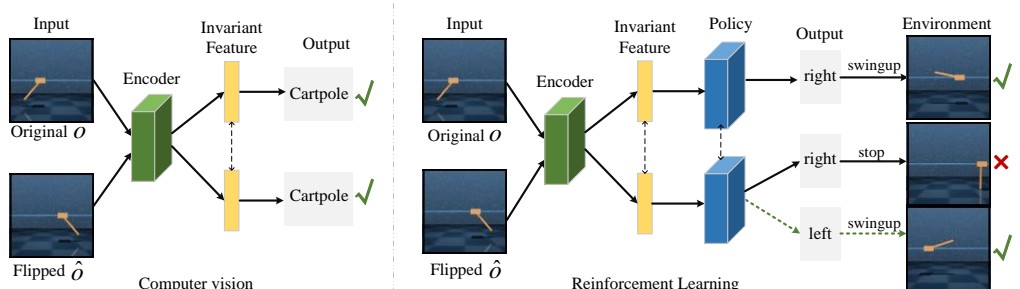

Figure 1: For image recognition task (left), the features of original images $o$ and flipped images $\hat{o}$ should be invariant for classification. In contrast, for cartpole swingup task (right) in RL whose goal is to swing up an unactuated pole, such invariant feature will guide policy network to output same actions for $o$ and $\hat{o}$. However, for $\hat{o}$, the better action is to apply reversed forces to this cart base to keep swingup. Thus the pre-collected action and reward for $o$ are not accurate for $\hat{o}$ any more.

applied on observations, the corresponding left and right actions should be reversed, thus collected rewards could not reflect the correct response.

Considering above problem, instead of working in the spatial domain, we aim to study the generalization of image-based RL from a frequency domain perspective. Our motivation stems from the recent studies Fourier analysis in computer vision [42]: different types of spatial corruptions influence the model robustness toward different frequency ranges. For example, the high frequency data augmentations such as random Gaussian noise bias the model towards utilizing low frequency information in the input, and improve robustness to corruptions with high frequencies. Moreover, single spatial-based data augmentation rarely improves model robustness of corruption types with different frequency characteristics. Inspired by this phenomenon, we conjecture that the observation's Fourier statistics for the same RL task also vary across different environment distributions. In Fig. 2, we visualize the observation's spectrum magnitude of the the original environment as well as the averaged delta between 4 shifted environment distributions on DeepMind Control suite [34]. The shifted environments show distinct spectrum difference with the original environment, with each one varying at different frequency regions. Knowing that a deep model often tends to bias on certain frequency band [39], we hypothesis that the diverse spectrum pattern across environment distributions is the key challenge of image-based RL generalization.

Based on the above analyses, we focus on the Frequency-based augmentation instead of the spatial-based one by introducing a Spectrum Random Masking (SRM) regularization, as shown in Fig. 3. Our proposed SRM can be easily compatible for most existing RL benchmarks. To prevent the model from focusing on a certain frequency range on input images, SRM randomly discards partial frequency of observations at training phase, and forces the policy to select the appropriate action with remaining information. In this way, we can increase the robustness to different corruptions (high, middle or low frequency characteristics) by considering the entire frequency distribution. Specifically, an input observation is operated by three steps: (1) the Fast Fourier transformation (FFT), (2) Spectrum random masking, and (3) the inverse Fourier transformation (IFFT), which enhances the diversity of training observations while maintaining the main content.

To summarize, this paper has the following contributions: (1) We provide a new frequency perspective for data augmentation in RL. In contrast to previous spatial-based methods, the proposed SRM are performed consistently for task-agnostic RL environments and could deal with diverse distribution by focusing on the whole frequency information. (2) The proposed SRM is a plug-and-play method that does not require any extra parameters or any additional observations from the environment. Moreover, it is complementary to existing data augmentation and regularization approaches. (3) We conduct comprehensive experiments to demonstrate SRM could bring performance gains on various image-based environments with high generalization potentials.

## 2  Background

**Reinforcement learning.** A RL problem is typically described as a Markov Decision Process (MDP) $\mathcal{M} =< \mathcal{S}, \mathcal{A}, \mathcal{P}, \mathcal{R}, \gamma >$, where $\mathcal{S}$ and $\mathcal{A}$ denotes the state space and action space respectively [33].

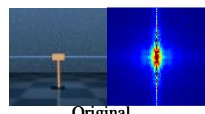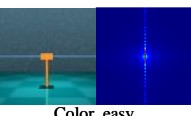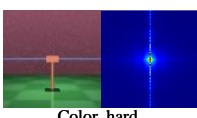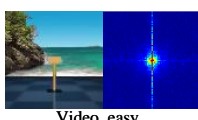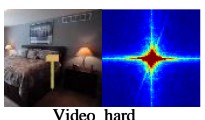

Original          Color_easy          Color_hard          Video_easy          Video_hard

Figure 2: Spectrum magnitude of the original environment observation $\mathbb{E}[\|\mathcal{F}(\mathcal{O})\|]$ and the observation spectrum magnitude difference $\mathbb{E}[\|\mathcal{F}(\mathcal{O} - \mathcal{D}(\mathcal{O}))\|]$ of four shifted environments. $\mathcal{F}$ is the 2D Fourier transform and $\mathcal{D}$ is the environment shift function. Sample observation images for each environment are placed on the left of corresponding spectrum map. A total of 500K frames on the cartpole swingup task in DeepMind Control suite are used for calculation and the results are averaged.

$\mathcal{P}(s_{t+1}|s_t, a_t)$ is the state transition, $R(s_t, a_t)$ is the reward function, and $\gamma \in [0, 1)$ is the discount factor. Here we consider a Partially Observable MDP (POMDP) [2] $\mathcal{M} = <\mathcal{O}, \mathcal{A}, \mathcal{P}, \mathcal{R}, \gamma>$ for the continuous control task because $\mathcal{S}$ is often not directly obtained from raw images, where $\mathcal{O}$ is the high-dimensional observation space of images. Following the convention [25], the POMDP is converted into an MDP by stacking several consecutive environment frames $\{o_t, o_{t-1}, o_{t-2}\}$ into a single state $s_t \in \mathcal{O}$.

The goal of typical RL is to find an optimal policy $\pi_\theta^*$ which maximizes the expected cumulative reward over the entire distribution of MDPs [33]:

$$J(\pi) = \sum_t \mathbb{E}_{(s_t, a_t) \sim \rho_\pi}[r(s_t, a_t)], \tag{1}$$

where $\rho_\pi(s_t, a_t)$ is the state-action marginal of the trajectory distribution. In this work, rather than learning an optimal policy on a single MDP, we consider a generalizable $\pi_\theta$, which is able to obtain high discounted return on a set of MDPs $\mathcal{M} = <\overline{\mathcal{O}}, \mathcal{A}, \mathcal{P}, \mathcal{R}, \gamma>$, where the observations in $\overline{\mathcal{O}}$ are unseen in training.

**Soft Actor-Critic.** Soft Actor-Critic (SAC) [9, 10] is a popular off-policy method for continuous control tasks. It learns a state-action value function $Q(s, a)$ and a stochastic policy $\pi(a|s)$ to find the optimal policy based on the maximum entropy RL framework:

$$J(\pi) = \sum_t \mathbb{E}_{(s_t, a_t) \sim \rho_\pi}[r(s_t, a_t) + \alpha\mathcal{H}(\pi(\cdot|s_t))], \tag{2}$$

where $\alpha$ is the temperature parameter which determines the relative importance of the entropy term against the reward. $\alpha = 0$ degenerates SAC into conventional RL. $\mathcal{H}(\cdot)$ provides a substantial improvement in exploration and robustness by enabling the probability of action output to be diverse as much as possible, rather than being concentrated on one action. Specially, $Q(s, a)$ is approximated by minimizing the soft Bellman residual:

$$J_Q = \mathbb{E}_{(s_t, a_t) \sim \mathcal{D}}(Q(s_t, a_t) - (r_t + \gamma\overline{V}(s_{t+1})))^2, \tag{3}$$

where $\mathcal{D}$ denotes the replay buffer, and $\overline{V}s_{t+1}$ is the soft target value network approximated as:

$$\overline{V}s_{t+1} = \mathbb{E}_{a_{t+1} \sim \pi}[\overline{Q}(s_{t+1}, a_{t+1}) - \alpha \log \pi(a_{t+1}|s_{t+1})], \tag{4}$$

where $\overline{Q}$ is the target $Q$ function, and the weights can be an exponentially moving average of the weights in $Q$. In the policy improvement step, the policy is updated by minimizing the divergence between the policy and the exponential of the soft Q function:

$$J_\pi = \mathbb{E}_{s_t \sim \mathcal{D}}\left[D_{KL}(\pi(\cdot|s_t) \Big\| \frac{\exp(Q(s_t, \cdot))}{Z(s_t)})\right]. \tag{5}$$

## 3 Methodology

### 3.1 Spectrum Random Masking

We firstly introduce how to generate new samples by spectrum random masking strategy in this section. Suppose we have a grayscale image observation [2] $o_i \in \mathbb{R}^{H \times W}$, the Fourier domain spectrum

---
[2]For RGB images, we apply the same operation separately on each color channel.

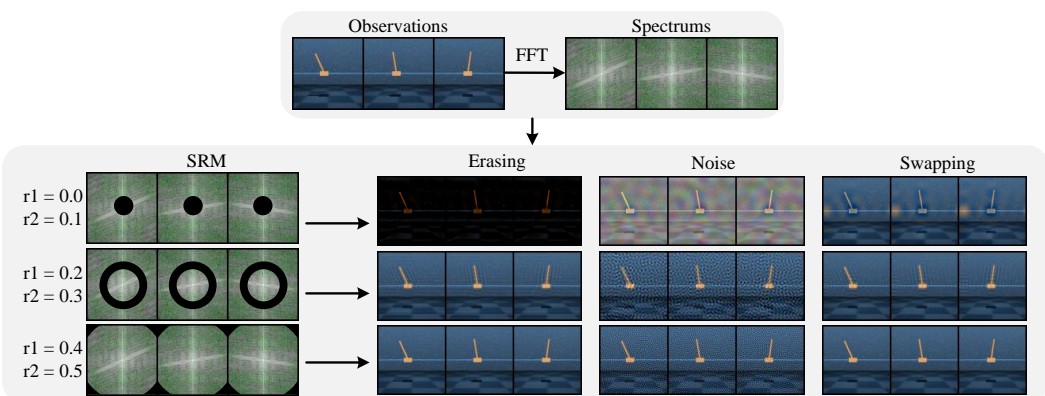

Figure 3: The framework and several examples of SRM. For three consecutive observations, SRM is adopted as data augmentation respectively. There are three types of masking strategies including directly erase, disturbed by noise or swapping. During training, the masking strategy, position and ratio could be dynamic changed.

$\mathcal{F}(o_i)$ can be calculated by fast Fourier transformation [26]:

$$\mathcal{F}(o_i)(u,v) = \Big( \sum_{h=0}^{H-1} \sum_{w=0}^{W-1} o_i(h,w)e^{-j2\pi(uh/H+vw/W)} \Big), \tag{6}$$

where $u \in [0, H-1]$ and $v \in [0, W-1]$ are horizontal and vertical indexes of frequency domain components, and $o_i(h,w)$ is the pixel value at position $(h,w)$. Note the subscript $i$ is just the index of the sample and irrelevant with the time step. For better processing and visualization, we use the shifted version of $\mathcal{F}(o_i)$ with the lowest frequency component placed in the center of the spectrum.

After transforming images from spatial to frequency domain, we construct a binary masking matrix $\mathbf{M} \in \{0,1\}^{H \times W}$. We introduce two radius parameters $r_1, r_2 \in [0, 0.5]$ to control the position and ratio of the spectral neighborhood to be masked[3] ($r_1 < r_2$). As shown in Fig. 3, $\mathbf{M}$ is constructed by setting the region in the annulus between two concentric circles (with radius $r_1$ and $r_2$) to 0. $\Delta r = r_2 - r_1$ decides the size of mask regions. The area ratio of masking region is defined as $r_p = r_2^2 - r_1^2$. Hence we propose 3 typical types of masking strategies: erasing, noise and swapping. If the frequency component lies between $r_1$ and $r_2$, it will be masked; otherwise, the frequency signals will be kept:

$$\hat{\mathcal{F}}(o_i) = \mathbf{M} \cdot \mathcal{F}(o_i) + (\mathbf{1} - \mathbf{M}) \cdot \mathcal{F}(\mathbf{Z}), \tag{7}$$

where $\mathbf{Z}$ is calculated by the specific masking strategy. For the erasing strategy, we set $\mathbf{Z}$ to $\mathbf{0}^{H \times W}$, and only the non-masked frequency components will be fully explored in follow-up training. The random noise strategy sets $\mathbf{Z}$ to a random noise image. The swapping strategy replaces the removed frequency components with the same frequency components from another image $\mathbf{I}$ by setting $\mathbf{Z} = \mathbf{I}$, which draws inspiration from Cutmix [43] and Mixup [47]. Considering that the third-party data $\mathbf{I}$ is not always available, during training, we randomly shuffle the training batch and pick $\mathbf{I} = o_j$ $(j \neq i)$ from the shuffled batch, where $o_j$ has the same batch index with $o_i$.

The masking strategy also influences the supervised signals since it is correlated with $o_i$. Suppose that $a_i, r_i$ is the supervised signals for observation $o_i$, where $a_i$ and $r_i$ stands for action and reward, respectively. For erasing and random noise strategy, $a_i$ and $r_i$ are kept unchanged. For the swapping strategy, the supervised signals depended on the area ratio of masking region $r_p$. If $r_p < 0.5$, we keep $a_i$ and $r_i$ unchanged, else we replace $a_i, r_i$ with $a_j$ and $r_j$.

Once the masked Fourier spectrum $\hat{\mathcal{F}}(o_i)(u,v)$ is obtained, we inverse shift and restore it to the spatial domain by the corresponding inverse Fourier transformation $\mathcal{F}^{-1}$:

$$\hat{o}_i(h,w) = \frac{1}{HW} \sum_{h=0}^{H-1} \sum_{w=0}^{W-1} \hat{\mathcal{F}}(o_i)(u,v)e^{j2\pi(uh/H+vw/W)}. \tag{8}$$

---

[3]Here we assume the diagonal length of the 2D Fourier spectrum is 1.0 to deal with different image sizes.

**Algorithm 1** Spectrum Random Masking

---

**Input**: Sample batch of transitions: $\{o_i, a_i, r_i, o'_i | i = 1, \cdots, K\} \sim \mathcal{B}$; Zero matrix $\mathbf{0} \in \mathbb{R}^{H \times W}$; Random noise image $\mathbf{N} \in \mathbb{R}^{H \times W}$; Masking matrix $\mathbf{M} \in \{0, 1\}^{H \times W}$.
**Output**: Augmented transitions $\{\hat{o}_i, \hat{a}_i, \hat{r}_i, \hat{o}'_i | i = 1, \cdots, K\}$
**Init**: $r_1 \leftarrow \texttt{Rand}(0, 0.5), r_2 \leftarrow \texttt{Rand}(r_1, 0.5), \Delta r = r_2 - r_1, r_p = r_2^2 - r_1^2$.

1: **for** $i = 1, \cdots, K$ **do**
2:     Fast Fourier transformation according to Eq. 6: $\mathcal{F}(o_i) \leftarrow o_i$
3:     **if** Masking strategies is 'Erasing' **then**
4:         $\hat{\mathcal{F}}(o_i) = \mathbf{M} \cdot \mathcal{F}(o_i) + (\mathbf{1} - \mathbf{M}) \cdot \mathcal{F}(\mathbf{0}), \hat{a}_i \leftarrow a_i, \hat{r}_i \leftarrow r_i$
5:     **else if** Masking strategies is 'Noise' **then**
6:         $\hat{\mathcal{F}}(o_i) = \mathbf{M} \cdot \mathcal{F}(o_i) + (\mathbf{1} - \mathbf{M}) \cdot \mathcal{F}(\mathbf{N}), \hat{a}_i \leftarrow a_i, \hat{r}_i \leftarrow r_i$
7:     **else if** Masking strategies is 'Swapping' **then**
8:         Random Shuffle Batch: $\{o_j, a_j, r_j, o'_j | j = 1, \cdots, K\}$
            $\hat{\mathcal{F}}(o_i) = \mathbf{M} \cdot \mathcal{F}(o_i) + (\mathbf{1} - \mathbf{M}) \cdot \mathcal{F}(o_j)$
9:         **if** $r_p \leq 0.5$ **then**
10:           $\hat{a}_i \leftarrow a_i, \hat{r}_i \leftarrow r_i$
11:         **else**
12:           $\hat{a}_i \leftarrow a_j, \hat{r}_i \leftarrow r_j$
13:         **end if**
14:     **end if**
15:     Inverse Fourier transformation according to Eq. 8: $\hat{o}_i \leftarrow \hat{\mathcal{F}}(o_i)$
16:     Execute Steps (2)$\sim$(15) with $o'_i$ to get $\hat{o}'_i$
17: **end for**

---

For observation $o_i$, its next observation $o_i\prime$ also goes through the same SRM operation as $o_i$. Algorithm 1 details how to generate new samples by SRM.

## 3.2 Reinforcement Learning with SRM

In off-policy RL, observations are sampled from a replay buffer. One intuitive way to use data augmentation in RL is to perform augmentation before passing them to the agent for training [41, 19]. Meanwhile, the augmentation procedure on stacked consecutive state frames $s_t = \{o_t, o_{t-1}, o_{t-2}\}$ should be consistent to keep temporal information (such as positions and velocities) invariant. Given an augmentation function $\tau(\cdot, v)$ with parameter $v \in \mathcal{V}$, the augmented observation $\hat{s}_t \in \hat{\mathcal{O}}$ of the original observation $s_t \in \mathcal{O}$ can be obtained by $\hat{s}_t = \tau(s_t, v)$. Optimally, $\tau : \mathcal{O} \times \mathcal{V} \to \hat{\mathcal{O}}$ should maintain Q-values and policy $\pi$ of the agent:

$$\begin{cases} Q(s_t, a_t) = Q(\tau(s_t, v), a_t), \\ \pi(a_t | s_t) = \pi(a_t | \tau(s_t, v)), \end{cases} \tag{9}$$

where $a_t \in \mathcal{A}$ is the action of $s_t$. However, practically it is unstable to learn value and policy functions even from the original observations, not to mention the augmented data that vary wildly in appearances. As a result, only several weak data augmentations such as random shift and random cropping could be employed in a plain way in RL optimization, as shown in DrQ [41] and RAD [19]. In contrast, strong data augmentations such as random convolution and rotation are always been applied in a specific way. In SVEA [13], both original and augmented images are used to optimize the Q-function, but only the former are adopted to compute Q-target of the Bellman equation. In SECANT [7], expert policy with weak augmentations guides student policy with strong augmentations for learning robust representations. Attractively, as we will show in the experiment section, SRM has the merit of both weak and strong data augmentations: it can be plugged into any image-based RL methods in a plain way, while providing significant performance improvement.

During training, we apply SRM with a probability of 0.5 on each batch of observations. No data augmentation is applied at test time. To evaluate the effectiveness of SRM, we adopt two popular methods DrQ [41] and SVEA [13] as our base algorithms. The former is a naive method where data augmentations are plugged into RL algorithms directly, and the corresponding SAC objectives are

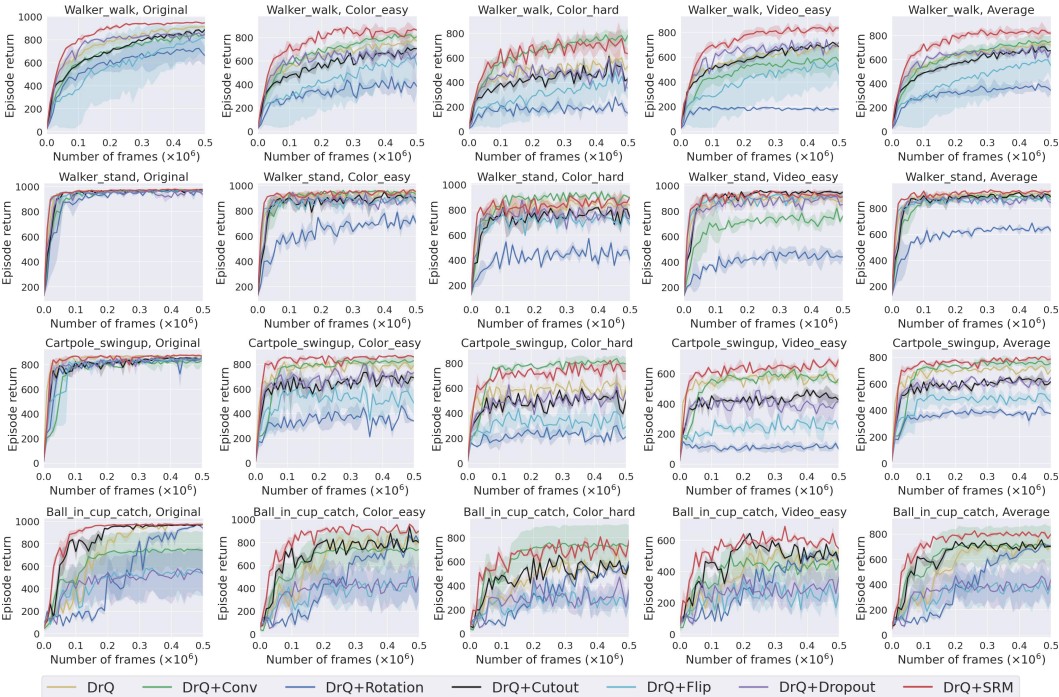

Figure 4: Comparisons of SRM with five data augmentation and one regularization methods. The average performance in the last column shows the generalization ability of SRM.

respectively replaced by:

$$J_Q = \mathbb{E}_{(s_t,a_t) \sim \mathcal{D}} \left[ \frac{(Q(s_t,a_t) + Q(\hat{s}_t, \hat{a}_t))}{2} - (r_t + \gamma \frac{\overline{V}(s_{t+1}) + \overline{V}(\hat{s}_{t+1})}{2})^2 \right]. \qquad (10)$$

The latter is delicately designed where Q-target is computed strictly from unaugmented data to alleviate the problem of high-variance, and its goal is:

$$J_Q = \mathbb{E}_{(s_t,a_t) \sim \mathcal{D}} \left[ \frac{(Q(s_t,a_t) + Q(\hat{s}_t, \hat{a}_t))}{2} - (r_t + \gamma \overline{V}(s_{t+1}))^2 \right]. \qquad (11)$$

By jointly solving the Q-function with policy and value network in Eqs. (4) and (5), the RL systems has the potential to learn from diverse observations for policy learning.

## 4 Experiments

### 4.1 Task Setup and Evaluation Protocol

**Environments** We conduct our experiment on 5 tasks from DeepMind Control Suite (DMControl) [34], which offers a set of challenging continuous control tasks and has been widely used as a common benchmark for visual RL [11, 1]. There are 2 kinds of observations in DMControl: state vector-based and image-based, and the latter is used in our experiment. It has been evidenced that the learning of state-based observations is easier than high dimensional image-based observations [18]. To evaluate the generalization, we train the agent from DMControl and report results tested on DMControl Generalization Benchmark (DMControl-GB) [14], which includes four distinct test distributions with slight/strong color/video attack: color-easy, color-hard, video-easy and video-hard..

**Implementation Details** For a fair comparison, we implement all methods following [13], where the same hyperparameters and network architecture are adopted. We use a 11-layer feed-forward convolution network as the shared encoder, which is followed by independent linear projections for the actor and critic. During training, the masking ratio and position of SRM are randomly chosen, and the ranges of $r_1$ and $\Delta r$ are set as $[0, 0.5]$ and $[0, 0.05]$ for each batch of observations, respectively. Note SRM is a data augmentation method and needs to combine with other learning baselines. To validate the effectiveness, 3 state-of-the-art baselines are employed: DrQ [41], SVEA-C [13] and

Table 1: Comparison with state-of-the-art methods on color-easy, color-hard, video-easy and video-hard benchmarks. `C` and `O` denotes random convolution and overlay respectively. The best results are in **bold**.

| Color, Easy | CURL | RAD | PAD | SODA-C | SODA-O | DrQ | DrQ +SRM | SVEA-C | SVEA-C +SRM | SVEA-O | SVEA-O +SRM |
|---|---|---|---|---|---|---|---|---|---|---|---|
| walker, walk | 645 ± 55 | 636 ± 33 | 687 ± 119 | - | 811 ± 41 | 826 ± 10 | 912 ± 21 | 758 ± 72 | **941 ± 20** | 868 ± 18 | 910 ± 26 |
| walker, stand | 866 ± 46 | 807 ± 67 | 894 ± 39 | - | 960 ± 4 | 929 ± 6 | 967 ± 3 | 832 ± 40 | 978 ± 4 | 974 ± 1 | **980 ± 1** |
| cartpole, swingup | 668 ± 74 | 763 ± 29 | 812 ± 20 | - | 859 ± 15 | 852 ± 15 | **878 ± 1** | 865 ± 3 | 871 ± 4 | 862 ± 8 | 867 ± 7 |
| ball_in_cup, catch | 565 ± 168 | 727 ± 87 | 775 ± 159 | - | 969 ± 3 | 840 ± 36 | 972 ± 4 | 969 ± 8 | **980 ± 1** | 976 ± 1 | 978 ± 2 |
| finger, spin | 781 ± 139 | 789 ± 160 | 870 ± 54 | - | 915 ± 43 | 861 ± 55 | 947 ± 21 | 904 ± 40 | 961 ± 20 | 963 ± 12 | **976 ± 14** |
| Average | 705 ± 96 | 744 ± 75 | 808 ± 78 | - | 903 ± 21 | 876 ± 42 | 935 ± 10 | 866 ± 33 | **946 ± 10** | 929 ± 8 | 942 ± 10 |
| Color, Hard | CURL | RAD | PAD | SODA-C | SODA-O | DrQ | DrQ +SRM | SVEA-C | SVEA-C +SRM | SVEA-O | SVEA-O +SRM |
| walker, walk | 445 ± 99 | 400 ± 61 | 468 ± 47 | 697 ± 66 | 692 ± 68 | 520 ± 91 | 806 ± 88 | 760 ± 145 | **907 ± 36** | 749 ± 61 | 836 ± 27 |
| walker, stand | 662 ± 54 | 644 ± 88 | 797 ± 46 | 930 ± 12 | 893 ± 12 | 770 ± 71 | 874 ± 14 | 942 ± 26 | **966 ± 11** | 933 ± 24 | 965 ± 13 |
| cartpole, swingup | 454 ± 110 | 590 ± 53 | 630 ± 63 | 831 ± 21 | 805 ± 28 | 586 ± 52 | 802 ± 28 | 837 ± 23 | **863 ± 15** | 832 ± 23 | 833 ± 24 |
| ball_in_cup, catch | 231 ± 92 | 541 ± 29 | 563 ± 50 | 892 ± 37 | 949 ± 19 | 365 ± 210 | 796 ± 36 | 961 ± 7 | **966 ± 12** | 959 ± 5 | 948 ± 27 |
| finger, spin | 691 ± 12 | 667 ± 154 | 803 ± 72 | 901 ± 51 | 793 ± 128 | 776 ± 134 | 889 ± 22 | **977 ± 5** | 974 ± 63 | 972 ± 6 | 971 ± 33 |
| Average | 497 ± 73 | 568 ± 77 | 652 ± 56 | 850 ± 37 | 826 ± 51 | 603 ± 112 | 833 ± 38 | 895 ± 41 | **935 ± 27** | 889 ± 24 | 911 ± 25 |
| Video Easy | CURL | RAD | PAD | SODA-C | SODA-O | DrQ | DrQ +SRM | SVEA-C | SVEA-C +SRM | SVEA-O | SVEA-O +SRM |
| walker, walk | 556 ± 133 | 606 ± 63 | 717 ± 79 | 635 ± 48 | 768 ± 38 | 682 ± 89 | 823 ± 32 | 618 ± 144 | 836 ± 40 | 819 ± 71 | **854 ± 42** |
| walker, stand | 852 ± 75 | 745 ± 146 | 935 ± 20 | 903 ± 56 | 955 ± 13 | 873 ± 83 | 947 ± 9 | 795 ± 70 | 932 ± 10 | 961 ± 8 | **966 ± 42** |
| cartpole, swingup | 404 ± 67 | 373 ± 72 | 521 ± 76 | 474 ± 143 | 758 ± 62 | 485 ± 105 | 740 ± 24 | 606 ± 85 | 808 ± 21 | 782 ± 27 | **812 ± 16** |
| ball_in_cup, catch | 316 ± 119 | 481 ± 26 | 436 ± 55 | 539 ± 111 | 875 ± 56 | 318 ± 157 | 651 ± 19 | 659 ± 110 | 882 ± 100 | 871 ± 106 | **924 ± 35** |
| finger, spin | 502 ± 19 | 400 ± 64 | 691 ± 80 | 363 ± 185 | 695 ± 97 | 533 ± 119 | 607 ± 92 | 764 ± 86 | 816 ± 93 | 808 ± 33 | **925 ± 4** |
| Average | 526 ± 83 | 521 ± 74 | 660 ± 62 | 583 ± 109 | 810 ± 53 | 578 ± 111 | 754 ± 35 | 688 ± 99 | 855 ± 53 | 848 ± 49 | **896 ± 28** |
| Video Hard | CURL | RAD | PAD | SODA-C | SODA-O | DrQ | DrQ +SRM | SVEA-C | SVEA-C +SRM | SVEA-O | SVEA-O +SRM |
| walker, walk | 58 ± 18 | 56 ± 9 | 93 ± 29 | - | 381 ± 72 | 104 ± 22 | 225 ± 29 | 224 ±7 | 364 ± 63 | 377 ± 93 | **535 ± 35** |
| walker, stand | 45 ± 5 | 231 ± 39 | 278 ± 72 | - | 771 ± 83 | 289 ± 49 | 325 ± 70 | 525 ±39 | 729 ± 29 | 834 ± 46 | **863 ± 57** |
| cartpole, swingup | 114 ± 15 | 110 ± 16 | 123 ± 24 | - | 492 ± 64 | 138 ± 9 | 254 ± 58 | 213 ± 26 | 425 ± 5 | 393 ± 45 | **523 ± 23** |
| ball_in_cup, catch | 115 ± 33 | 97 ± 29 | 66 ± 61 | - | 327 ± 100 | 92 ± 23 | 237 ± 6 | 413 ±57 | 339 ± 17 | 403 ± 174 | **566 ± 135** |
| finger, spin | 27 ± 21 | 34 ± 11 | 56 ± 18 | - | 302 ± 41 | 71 ± 45 | 92 ± 36 | **430 ±44** | 335 ± 21 | 335 ± 58 | 419 ± 32 |
| Average | 72 ± 18 | 106 ± 21 | 123 ± 41 | - | 455 ± 72 | 139 ± 30 | 227 ± 40 | 361 ±35 | 438 ± 27 | 468 ± 83 | **581 ± 56** |

SVEA-O [13], denoted as "DrQ+SRM", "SVEA-C+SRM" and "SVEA-O+SRM" respectively. O and C represent random convolution and random overlay respectively, which are set as the default augmentation used in SVEA [13]. For DrQ, SVEA-C and SVEA-O, the better performances between the reported ones of the original paper and our re-run ones using the source codes are reported in our experiment. We provide 3 candidate masking strategies erasing, noise and swapping in Algorithm 1, all of them can achieve higher performance than baseline as shown in Table 5 . Since erasing performs better than others, it is chosen as the default masking strategy in the following experiments if not otherwise mentioned. All methods are trained for 500K frames. We run 5 times with random seeds and report the averaged results.

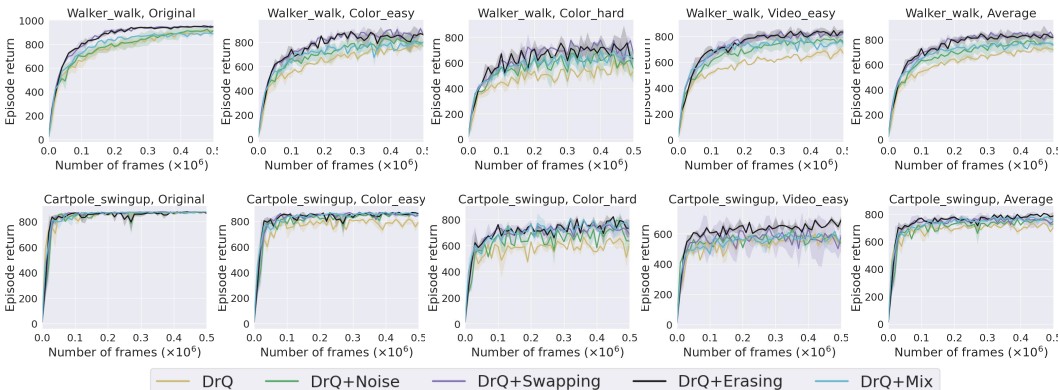

Figure 5: The results of different masking strategies. Either way, disturbed in frequency domain is a useful way to improvement performance.

## 4.2 Comparisons

**Comparisons with state-of-the-arts** The comparisons are exhibited in Table 1. For the direct comparison, "DrQ+SRM", "SVEA-C+SRM" and "SVEA-O+SRM" outperform DrQ, SVEA-C and SVEA-O with a clear margin, respectively. In addition, the usage of SRM reduces the standard deviation and shows the strong leaning stability. It indicates that performing augmentation in the frequency domain is effective for image-based RL tasks. In addition, the performance of other methods such as CURL [18], PAD [12], RAD [19] and SODA [14] are also reported. Since our method is implemented with different learning baselines with them, the results are just provided for reference. Overall, it is evident that SRM provides new state-of-the-art performance for the task.

**Comparisons with spatial-based data augmentation** Here we compare SRM with 5 spatial-based data augmentation methods and 1 removal regularization strategy, including random convolution [20], random rotation [8], random cutout [5], random flip [19] and dropout [31]. DrQ is employed as the baseline learning method. When utilizing each data augmentation method, 50% of frames are remain un-augmented. This is because we find that directly using all kinds of data augmentation is harmful to policy stability. The dropout layer is added after every nonlinearity layer in the MLP of actor and critic networks and the dropout probability is set to 0.1.

Since the video hard task is extremely hard and the learning of DrQ is oscillating, we show learning curves of the different data augmentations in the benchmarks of color-easy, color-hard and video-easy. As shown in Fig. 4, the spatial data augmentation methods are less effective and even degrade performance, especially for flip and rotation. The reason is that these augmentations inevitably changes the environment dynamics, leading to inaccurate state transitions. In contrast, SRM outperforms the baseline in most scenarios and achieve comparable results in the rest scenarios. It demonstrates that the generalization ability of frequency based SRM is stronger than its spatial-based counterparts. Note that the "random cov" performs well on color hard and is less effective than SRM in other situations. The reason is that applying random convolution mainly changes the low frequency components (i.e., colors and textures) of an image, which makes the agent robust to color changes. However, when facing environments that differs significantly on higher frequency components, random convolution becomes less effective. Compared with "random cov", SRM works more generally to different scenarios.

Fig. 4 also tells that, although the removal based methods dropout and cutout has been proved to be able to improve generalization for discriminative models, they decrease the performance of RL tasks. For cutout, randomly cutting a region of an input image is prone to information loss, especially when the target region (e.g., the cartpole in cartpole swingup task) is removed. The performance decrease of dropout can be attributed to the chaotic nature of the RL training when facing with random dropped hidden units. In contrast, each frequency component in the Fourier spectrum has the global vision of the whole image, thus SRM does not remove critical image regions. Meanwhile, as an augmentation method, SRM also does not influence hidden units of networks.

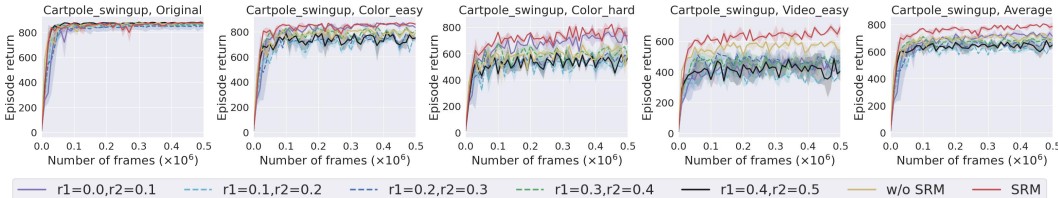

Figure 6: The influence of masking position. Masking frequency on a random band is better than on a fixed band.

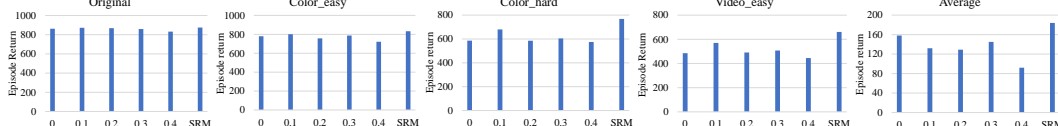

Figure 7: The influence of masking ratio. Masking frequency within a random ratio is better than within a fixed ratio.

### 4.3 Ablation Study

**Spectrum Masking Strategies** There are four different frequency masking strategies including erasing, noise, swapping and a mixture of the first three. As shown in Fig. 5, we study the choice of masking value on the performance of RL and have the following observations: First, all of these masking strategies are able to improve the average performance. This benefits from the mechanism of random masking which attempt to strengthen the whole frequency learning while suppress overfitting to certain frequency range. This also demonstrates that the agent may underfit to certain frequency range and overfit to other range when directly training with original observations. Second, with the increasing complexity of frequency disturbance, the performance of mixture is sometimes much worse than solely using erasing or swapping. This phenomenon can be attributed to introduce more variance by mixture.

**Spectrum Masking Position** To explore how the low and high frequency affect policy training, we set $r_1$ to 0.1, 0.2, 0.3, 0.4 respectively, and $\Delta r$ to 0.1. According to Frequency Principle (F-Principle), DNN tends to captures the frequency from low to high components in ascending complexity order. This is consistent with our results as shown in Fig. 6. At the early training stage, masking low frequency lead to slower improvement, while masking high frequency leads to a similar improvement with the non-SRM baseline. In the later period, remaining high frequency is able to achieve higher results than remaining low frequency. Moreover, SRM could get the best results in most benchmarks than fixed masking position. It demonstrates that learning frequency from low to high uniformly in a random way is beneficial.

**Spectrum Masking Ratio** To better control the masking ratio, we set $\triangle r$ to 0.05, 0.1,0.2,0.3 and 0.4 respectively. The results are shown in Fig. 7. Notably, SRM with random masking ratio consistently outperforms the DrQ baseline under all parameters setting, especially for harder environment. In addition, we can observe that masking ratio has little influence on original environment, which can be attributed to the remaining of a decent percentage of un-augmented frames.

## 5  Related works

Generalization in RL draws increasing attention in recent years. Existing methods can be roughly categorized into two groups: (1) increasing distribution similarity between training and testing environments by augmentation methods (such as data augmentation and environment generation) [16, 20, 41, 18, 19, 14, 28] and (2) handling differences between the training and testing MDPs (such as regularisation) [15, 22, 45, 46, 6, 1, 21, 36, 23, 29, 37]. Our method belongs to the first group but works in the frequency domain, which is ignored by most current RL methods. To explain the generalization behaviors of CNN, recent works in computer vision also provide new insights from the frequency domain aspect [39, 35, 42, 40, 38, 3]. However, they mainly consider the domain shift between fixed datasets on discrimination tasks (e.g., classification and segmentation). In contrast, we consider the dynamic environment in RL as well as the joint representation learning and dedecision-making process. Moreover, CNN tendssince CNN tends to rely on either low frequency

or high frequency components for fixed tasks during training [39], existing methods rarely consider regularizing the model on all frequency ranges. Differently, our method considers the broad frequency difference between different RL environment distributions and regularizes the model on all frequency ranges uniformly.

## 6 Conclusions

In this work, we investigated the generalization of image-based RL from a frequency domain perspective. Motivated by the distinct spectral impacts of different environment distribution shifts, we develop Spectrum Random Masking (SRM), a simple yet effective frequency domain augmentation methodwhich randomly discards certain frequency bands of the observation. Experimental results show that SRM improves model robustness under various distribution shifts without violating the environment dynamics. The effectiveness of SRM highlights a promising future research direction for researchers to take into account the frequency domain intrinsic mechanism of RL models. As RL tasks are essentially different from conventional computer vision tasks (e.g., image recognition), we believe that different conclusions will be made to provide further intuition and benefit the model performance.

Limitations: As an image-based augmentation method, our proposed SRM resort to enhancing the diversity of image-base observations while maintaining the crucial reward-related content, which facing challenges when image observations are incomplete or only state information can be observed.

## 7 Acknowledgments and Disclosure of Funding

This work is partially supported by grants from the Key-Area Research and Development Program of Guangdong Province under contact No. 2020B0101380001 and grants from the National Natural Science Foundation of China under contract No. 62027804, No. 61825101 and No. 62088102. The computing resources of Pengcheng Cloudbrain are used in this research.

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
