# Supplementary Materials of "Spectrum Random Masking for Generalization in Image-based Reinforcement Learning"

**Yangru Huang**[1], **Peixi Peng**[1 2 *], **Yifan Zhao**[1], **Guangyao Chen**[1], **Yonghong Tian**[1 2 *]

[1]School of Computer Science, Peking University, Beijing, China
[2]Peng Cheng Laboratory, ShenZhen, China

The sections of this appendix are organized as follows:

- Section A describes the details of experimental setup including network architecture, hyper-parameters, computing infrastructure and implementation code of SRM.

- Section B provides additional experimental results including the direct performance gain of SRM on SAC, the impacts of spatial and spectral masking, the effectiveness of various alternative spectral mask forms and the performance on video-hard benchmark.

- Section C demonstrates the effectiveness of SRM in more challenging and realistic environments, including DrawerWorld [15] (the background with realistic textures), Robosuite [21] (both target and background with progressively harder textures and color), and CARLA [2] (a driving simulator including various weathers).

- Section D gives detailed introduction and comparison of image-based RL algorithms, data augmentations for generalization, and several recently proposed masked image/frequency modeling methods related to SRM.

- Section E analyses the limitation of current work and suggest an interesting direction (generalization towards viewpoint changes) for future studies.

- Section F states the potential negative societal impacts.

## A  Detailed Experimental Setup

**Network architecture** Following the experiments in [6] and [7], the network architecture for DMControl includes a shared encoder, an actor subnet and a critic subnet, as illustrated in Fig. 1. The shared encoder $f_\theta$ consists of 11 convolutional layers with kernel size set to $3 \times 3$. The first convolution layer has a stride of 2. All the rest convolution layers are followed by a ReLU layer and the corresponding stride is set to 1. The encoder takes a stack of $3 \times 84 \times 84$ RGB frames as input and outputs $32 \times 21 \times 21$ feature maps, where the tensor dimensions are in the order of the number of channels, height and width, respectively. The $32 \times 21 \times 21$ feature maps are used as input to the actor and critic subnets, which are implemented by independent linear projections. Specifically, the projections consist of 3 linear layers with hidden dimension 1024, where the first two layer is followed by a ReLU layer. For the critic subnet, the input also includes corresponding actions, and the output is Q-value. The actor subnet outputs the mean and standard deviation of a Gaussian probability distribution for the continuous action space, and randomly selects actions based on the distribution.

**Hyper-parameters** The related hyper-parameters of our experiments are detailed in Table 1. For a fair comparison, we adopt the same hyperparameters as [7].

**Computing infrastructure** We train all models with a server, which is equipped with NVIDIA GeForce RTX 3080 GPUs and a 256 core AMD EPY 7H12 2.6GHz CPU Processor.

---

*Corresponding author

36th Conference on Neural Information Processing Systems (NeurIPS 2022).

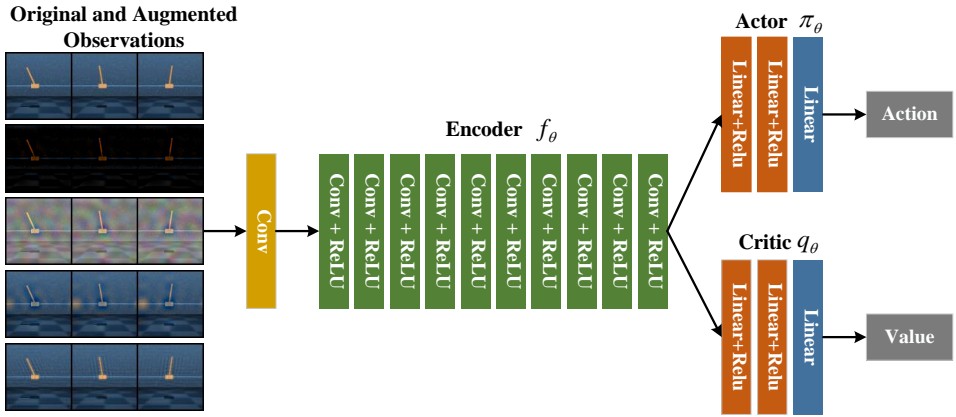

Figure 1: Network architecture on the DMControl tasks includes three parts: a shared encoder, a actor subnet and a critic subnet.

Table 1: Hyper-parameters used in the DMControl tasks.

| Hyperparameter | Value |
|---|---|
| Masking strategies | [Erasing, Noise, Swapping] |
| $r_1$ | [0,0.5] |
| $\Delta r$ | [0,0.05] |
| Frame rendering | $3 \times 84 \times 84$ |
| Stacked frames | 3 |
| Action repeat | 2(finger) |
| | 8(cartpole) |
| | 2(otherwise) |
| Discount factor $\lambda$ | 0.99 |
| Episode length | 1,000 |
| Learning algorithm | Soft Actor-Critic (SAC) |
| Number of frames | 500,000 |
| Replay buffer size | 500,000 |
| Optimizer $(f, \pi, q)$ | Adam $(\beta_1 = 0.9, \beta_2 = 0.999)$ |
| Optimizer $(\alpha$ in SAC) | Adam $(\beta_1 = 0.5, \beta_2 = 0.999)$ |
| Learning rate $(f, \pi, q)$ | 1e-3 |
| Learning rate $(\alpha$ in SAC) | 1e-4 |
| Batch size | 128 |
| Update frequence $(\theta)$ | 2 |

**Code** We provide PyTorch implementation for SRM-Erasing, as shown in Table 2. The implementations of SRM-Noise and SRM-Swapping are similar to SRM-Erasing. We only need to change `out_spectrum = x_spectrum * M` to `out_spectrum = x_spectrum * M + y_specturm * (1-M)`, where `y` is a noise image or a shuffled image for SRM-Noise and SRM-Swapping respectively.

## B   Additional Experimental Results on DMControl

**The directly performance gain by SRM on SAC** In the body, to provide a result of SOTA, we adopt DrQ [18] and SVEA [7] as baselines respectively. However, both of them apply random shift as default data augmentation method. To analyse the performance gain by SRM itself, we remove spatial-based data augmentation random shift in DrQ [18], which means that SAC [5] is adopted as baseline. During training, both original observations and generated observations by SRM are used to caculate Q-target and Q-function like DrQ. As shown in Figure 2, for 8 out 9 selected tasks, the test results of using SRM are much higher than baselines on 5 shifted distributions. This demonstrates that the proposed SRM itself is able to facilitate learning a universal policy to handle various distribution shifts.

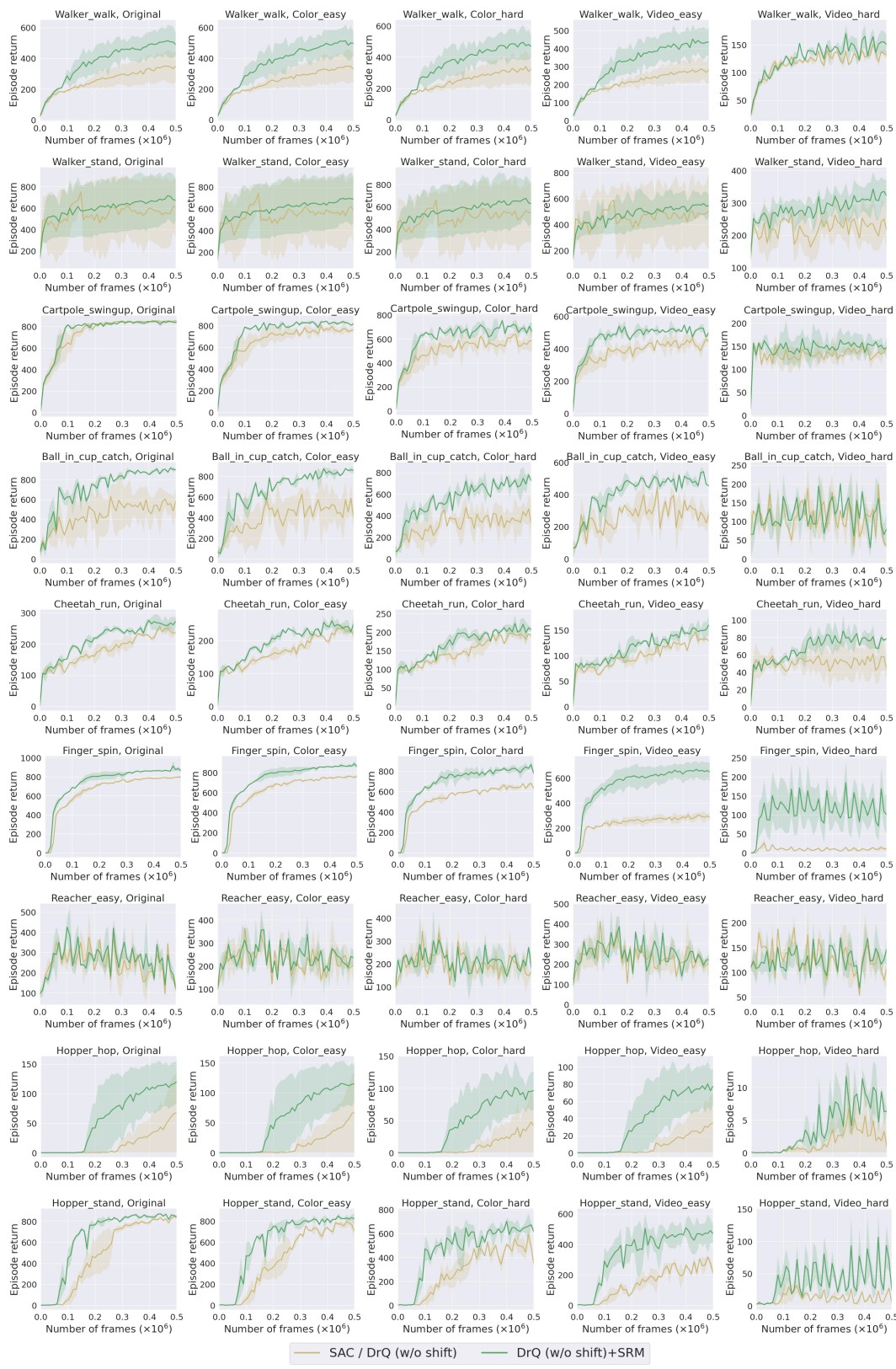

Figure 2: The directly performance gain by SRM on SAC with 9 different DMControl tasks.

Table 2: The code of SRM-Erasing.

```python
import torch
import numpy as np
def SRM_erasing(x):
    r1 = random.uniform(0,0.5)
    delta_r = random.uniform(0,0.05)
    r2 = np.min((r1 + delta_r, 0.5))
    # generate Mask M
    B,C,H,W = x.shape
    center = (int(H/2), int(W/2))
    diagonal_lenth = np.sqrt(H**2+W**2)
    r1_pix = diagonal_lenth * r1
    r2_pix = diagonal_lenth * r2
    Y_coord, X_coord = np.ogrid[:H, :W]
    dist_from_center = np.sqrt((Y_coord - center[0])**2 + (X_coord - center[1])**2)
    M = dist_from_center <= r2_pix
    M = M * (dist_from_center >= r1_pix)
    M = ~ M
    # mask Fourier spectrum
    M = torch.from_numpy(M).float().to(x.device)
    srm_out = torch.zeros_like(x)
    for i in range(C):
        x_c = x[:,i,:,:]
        x_spectrum = torch.fft.fftn(x_c, dim=(-2,-1))
        x_spectrum = torch.fft.fftshift(x_spectrum, dim=(-2,-1))
        out_spectrum = x_spectrum * M
        out_spectrum = torch.fft.ifftshift(out_spectrum, dim=(-2,-1))
        srm_out[:,i,:,:]  = torch.fft.ifftn(out_spectrum, dim=(-2,-1)).float()
    return srm_out
```

Table 3: The impacts of various mask forms.

| Task\Mask | DrQ | SRM | FAN_45 | FAN_90 | FAN_135 | FAN_180 | BandPass | LowPass | HighPass |
|---|---|---|---|---|---|---|---|---|---|
| color_easy | 826±10 | 912±21 | 634±24 | 710±17 | 645±23 | 775±32 | 562±27 | 818±34 | 150±20 |
| color_hard | 520±91 | 806±88 | 384±74 | 559±71 | 491±87 | 588±94 | 463±62 | 532±68 | 136±59 |
| video_easy | 682±89 | 823±32 | 685±94 | 586±55 | 657±79 | 785±68 | 502±77 | 674±87 | 113±63 |
| video_hard | 104±22 | 225±29 | 119±16 | 121±25 | 185±17 | 225±20 | 99±14 | 62±19 | 56±17 |

**Comparison between spatial masking and spectral masking** Spatial masking and spectral masking is operated in the pixel and frequency domain respectively. Similar to SRM, spatial masking also has three strategies, including Cutout, Cut-Noise and Cutmix. As shown in Fig. 3, Fig. 4 and Fig. 5, spectral masking is demonstrated superior over spatial masking on walker-walk task and cartpole-swingup task. We conjecture the main reason is that the task-related part could be discarded with a certain probability in spatial-based masking, leading to the confusion of RL agent.

**Comparison between different mask forms** In SRM we use ring-shaped masks for spectral information erasing. To evaluate the impacts of various different alternative mask forms, we conduct experiments on DMControl walker_walk task using fan-shaped masks with different angles, high-pass filters, low-pass filters, and band-pass filters. DrQ are adopted as the base algorithm. The results are shown in Table 3. As we can see, fan-shaped masks generally perform worse than SRM. The reason might be that masking a fan-shaped section drops too much information and cannot eliminate any frequency band entirely at the same time. The performance of increases with larger fan angles, indicating that it is better to eliminate the entire frequency band ring. The three band masks also perform worse than SRM, indicating that it is better to mask a wide range of bands instead of leaving only a certain band.

**The results on video-hard benchmark** Since the video-hard task is extremely hard and the learning of DrQ is oscillating, we report the related results on the better baselines SVEA_Conv and SVEA_Overlay, as shown in Fig. 6. As we can see, SVEA_Conv+SRM and SVEA_Overlay+SRM could achieve higher performance than SVEA_Conv and SVEA_Overlay on 4 different tasks. The phenomenon demonstrate that, when other spatial-based data augmentation are combined with SRM, the episode return could be further improved. We infer that SRM may help spatial-based augmentation play a greater role.

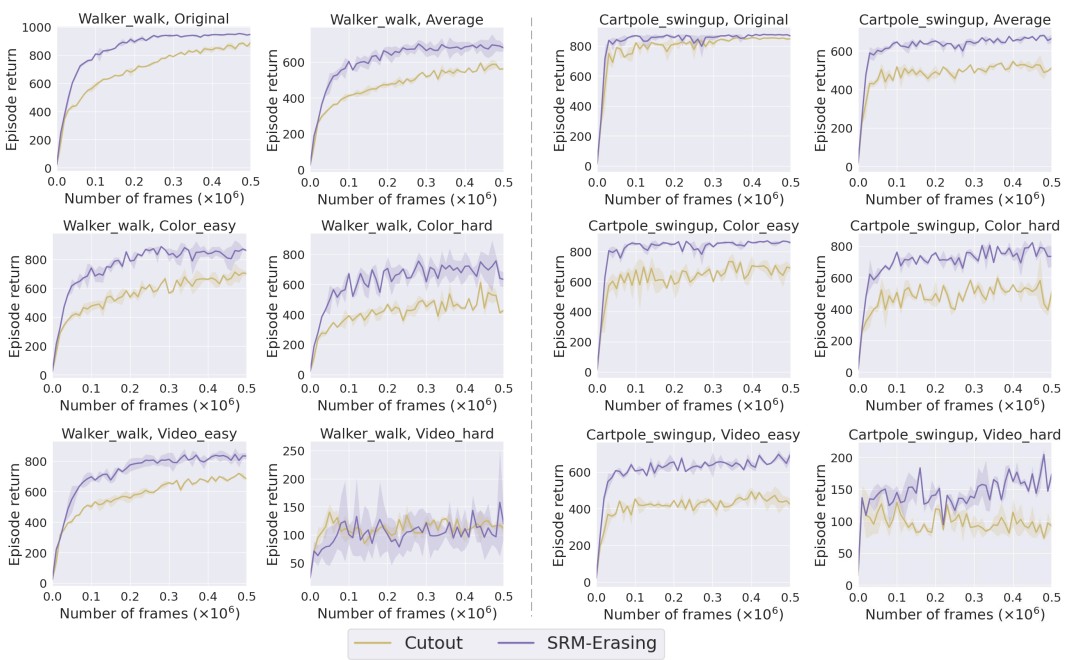

Figure 3: Comparison between Cutout (spatial-based) and SRM-Erasing (spectral-based).

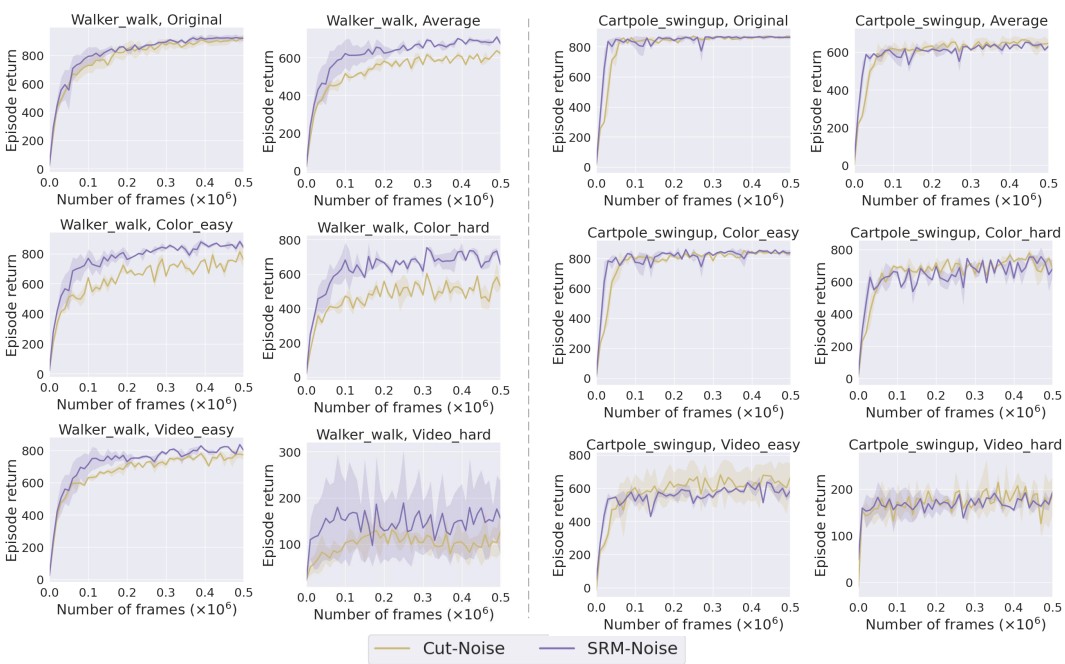

Figure 4: Comparison between Cut-Noise (spatial-based) and SRM-Noise (spectral-based).

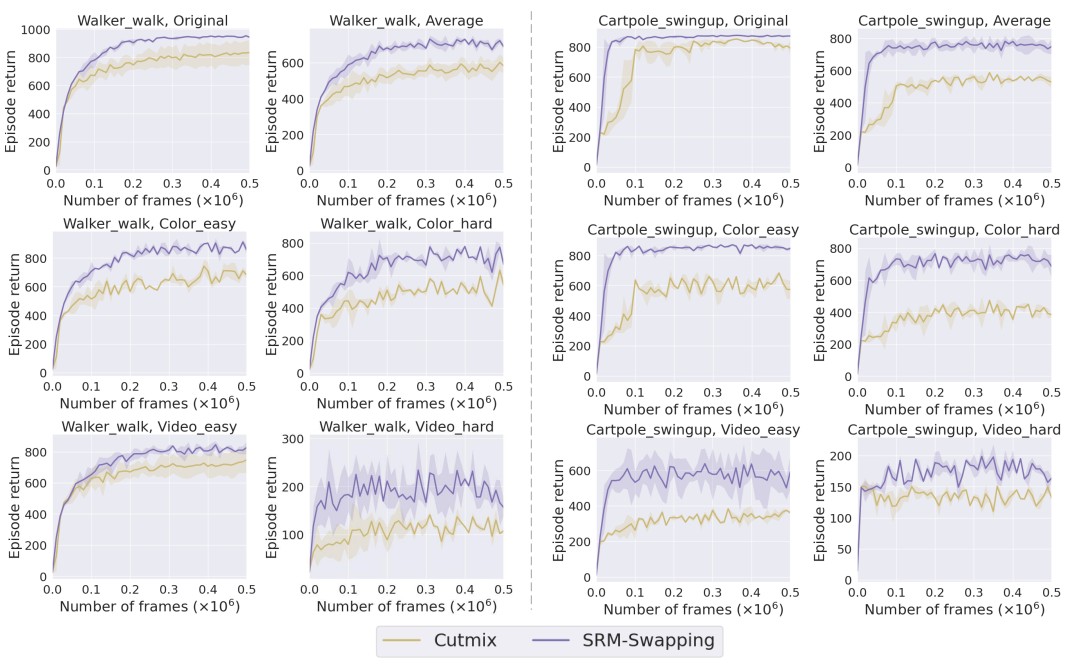

Figure 5: Comparison between Cutmix (spatial-based) and SRM-Swapping (spectral-based).

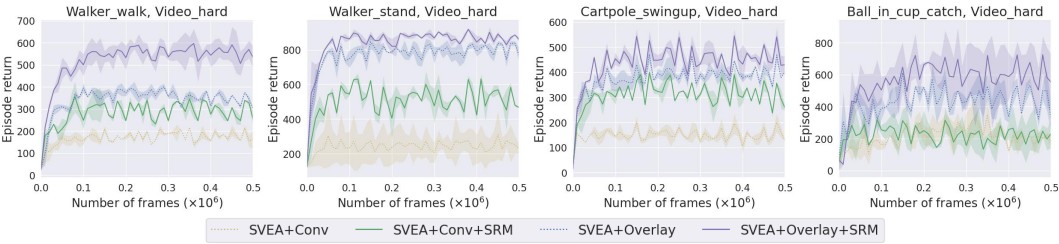

Figure 6: The test results on video-hard benchmark where SRM are combined with random convolution or random overlay during training.

## C  Additional Benchmarks on More Realistic Environments

To further evaluate the generality of SRM in more challenging and realistic environments, we provide experiments on additional 3 different benchmarks DrawerWorld, CARLA and Robosuite. DrawerWorld is a realistic texture benchmark for robot manipulation. Robosuite is a robitic simulator with more distracting textures of the table, floor, and objects. CARLA is a driving simulator including various weathers. All results are reported by running five times. We set the related hyper-parameters of DrawerWorld, CARLA and Robosuite following [15, 3]

### C.1  DrawerWorld

DrawerWorld (a variant of MetaWorld [20] with visual observations) is a realistic texture benchmark for manipulation proposed by [15]. Here we conduct experiments on DrawerOpen and DrawerClose tasks where a Sawyer arm is manipulated to open a drawer. The agents are trained on the grid texture and tested on 6 different realistic textures including grid, black, blanket, fabric, metal, marble, and wood. The results are shown in Table 4. The success rates of the DrQ baseline is significantly boosted with SRM under different background textures, which demonstrates its effectiveness. We also achieve the second best performance among all methods (The best model VAI [15] uses sophisticated keypoint detection and foreground restoration networks to remove the background distraction completely.

Table 4: The Success Rate (%) of different methods on DrawerOpen and DrawerClose tasks of DrawerWorld.

| | DrawerOpen | | | | | | |
|---|---|---|---|---|---|---|---|
| Texture\Method | SAC | PAD | VAI | DrQ | DrQ+SRM | SVEA_C | SVEA_C+SRM |
| Grid | 98±2 | 84±7 | 100±0 | 82±3 | 89±4 | 92±4 | 95±2 |
| Black | 95±2 | 95±3 | 100±1 | 75±7 | 91±5 | 87±8 | 93±3 |
| Fabric | 2±1 | 20±6 | 99±1 | 25±4 | 56±5 | 61±5 | 70±7 |
| Metal | 35±7 | 81±3 | 98±2 | 79±5 | 92±2 | 70±7 | 88±4 |
| Wood | 18±5 | 39±9 | 94±4 | 35±7 | 72±5 | 43±9 | 73±4 |
| Average | 49.6 | 63.8 | 98.2 | 59.2 | 80 | 70.6 | 83.8 |
| | DrawerClose | | | | | | |
| Grid | 100±0 | 95 ±3 | 99±1 | 91±3 | 93±5 | 93±6 | 96±4 |
| Black | 75 ±4 | 64 ±9 | 100 ±0 | 74±12 | 83±10 | 76±23 | 89±18 |
| Fabric | 0±0 | 0±0 | 74±8 | 8±6 | 17±7 | 24±15 | 38±17 |
| Metal | 0±0 | 2±2 | 98±3 | 11±10 | 29±8 | 27±18 | 39±12 |
| Wood | 0±0 | 12±2 | 70±6 | 10±17 | 22±15 | 25±21 | 37±14 |
| Average | 25 | 25 | 82 | 38.8 | 48.8 | 49 | 59.8 |

Table 5: Results on CARLA. The environment with clear noon are used for training and other weathers with the changes of shadow, sunlight and raining are utilized for testing. The travelled distances (m) in a town without collision are reported.

| Weather | SECANT | SAC | SAC+crop | DR | NetRand | SAC+IDM | PAD | DrQ | DrQ+SRM | SVEA_C | SVEA_C+SRM |
|---|---|---|---|---|---|---|---|---|---|---|---|
| Clear noon | 596 ±77 | 282 ±71 | 684 ±114 | 486 ±141 | 648 ±61 | 582 ±96 | 632 ±126 | 586 ±108 | 671 ±149 | 616 ±97 | 675 ±146 |
| Wet sunset | 397 ±99 | 57 ±14 | 26 ±18 | 9 ±11 | 284 ±84 | 25 ±11 | 36 ±12 | 34 ±15 | 93 ±21 | 297 ±121 | 354 ±95 |
| Wet cloudy noon | 629 ±204 | 180 ±45 | 283 ±85 | 595 ±260 | 557 ±107 | 433 ±105 | 515 ±52 | 393 ±89 | 490 ±92 | 519 ±112 | 576 ±108 |
| Soft rain sunset | 435 ±66 | 55 ±28 | 38 ±25 | 25 ±41 | 251 ±104 | 36 ±32 | 41 ±37 | 97 ±67 | 160 ±82 | 208 ±103 | 239 ±93 |
| Mid rain sunset | 470 ±202 | 50 ±8 | 37 ±16 | 24 ±24 | 233 ±117 | 42 ±23 | 32 ±21 | 76 ±48 | 136 ±78 | 205 ±105 | 240 ±85 |
| Hard rain noon | 541 ±96 | 237 ±85 | 235 ±129 | 341 ±96 | 458 ±72 | 156 ±194 | 308 ±141 | 289 ±93 | 342 ±84 | 429 ±83 | 461 ±94 |

In contrast, our SRM is a plug-and-play data augmentation method without any additional model modification. Hence, the comparison with VAI is just listed for reference.) Specially, we boost the performance of DrQ by +35.13% and +25.77% on two tasks, respectively.

## C.2 CARLA

CARLA [2] is a driving simulator including various weathers with highly realistic raining, shadow, and sunlight changes. Here we train the agents on clear noon weather and evaluate them on clear noon, wet sunset, wet cloudy noon, soft rain sunset, mid rain sunset and hard rain noon. The average distance travelled in the Town04 without collision are reported over 10 episodes per weather. As shown in Table 5, SRM is able to improves the baseline models' generalization ability on CARLA. Specially, using SRM can increase the driving distance of an autonomous driving car by +27.4% and +11.9% with DrQ and SVEA-C as baseline methods, respectively.

## C.3 Robosuite

Robosuite [21] is a robitic modular simulator. Here we mainly test the generalization of SRM on the scenes with more distracting textures of the table, floor, and objects with the Franka Panda robot. Clean background and objects are used during training and three sets of environments with progressive difficulty are adopted for testing. We follow the training settings introduced in [3]. As shown in Table 6, SRM boosts baseline methods on Robosuite and achieves better average performance than other methods except for SECANT. SECANT uses expert policy with weak augmentation data to guide student policy with strong augmentation data, which is orthogonal to our SRM. Hence, the comparison with SECANT is just listed for reference. In theory, SRM could be seamlessly integrated into SECANT for further performance gain. Specifically, on Robosuite (please refer to the second table in our general response for all reviewers), adding our proposed SRM on DrQ (denoted as DrQ+SRM) can boost the performance of DrQ by +169%, +560%, +550% on the DoorOpening

Table 6: Results on Robosuite. Clean background and objects are used for training and three sets of environments with progressive difficulty are adopted for testing.

| Hardness | Tasks\Methods | SECANT | SAC | SAC+crop | DR | NetRand | SAC+IDM | PAD | DrQ | DrQ+SRM | SVEA_C | SVEA_C+SRM |
|---|---|---|---|---|---|---|---|---|---|---|---|---|
| Easy | Door opening | 782 ±93 | 17 ±12 | 10 ±8 | 177 ±163 | 438 ±157 | 3 ±2 | 2 ±1 | 79 ±52 | 213 ±105 | 325 ±184 | 452 ±112 |
|  | Nut assembly | 419 ±63 | 3 ±2 | 6 ±5 | 12 ±7 | 242 ±28 | 13 ±12 | 11 ±10 | 16 ±12 | 87 ±27 | 150 ±31 | 251 ±24 |
|  | Two-arm lifting | 610 ±28 | 29 ±11 | 23 ±10 | 41 ±9 | 62 ±43 | 20 ±8 | 22 ±7 | 24 ±10 | 45 ±11 | 49 ±37 | 81 ±34 |
|  | Peg-in-hole | 837 ±42 | 186 ±62 | 134 ±72 | 139 ±37 | 390 ±68 | 150 ±41 | 142 ±37 | 179 ±61 | 262 ±54 | 301 ±67 | 335 ±58 |
| Hard | Door opening | 522 ±131 | 11 ±10 | 11 ±7 | 37 ±31 | 133 ±82 | 2 ±1 | 2 ±1 | 15 ±8 | 99 ±75 | 102 ±43 | 154 ±37 |
|  | Nut assembly | 437 ±102 | 6 ±7 | 9 ±8 | 33 ±18 | 181 ±53 | 34 ±28 | 24 ±26 | 21 ±15 | 49 ±14 | 74 ±28 | 167 ±47 |
|  | Two-arm lifting | 624 ±40 | 28 ±11 | 27 ±9 | 61 ±15 | 41 ±25 | 17 ±6 | 19 ±8 | 24 ±12 | 57 ±10 | 47 ±19 | 69 ±27 |
|  | Peg-in-hole | 774 ±76 | 204 ±81 | 143 ±62 | 194 ±41 | 322 ±72 | 165 ±75 | 164 ±69 | 162 ±58 | 231 ±67 | 260 ±81 | 328 ±75 |
| Extreme | Door opening | 309 ±147 | 11 ±10 | 6 ±4 | 52 ±46 | 140 ±107 | 2 ±1 | 2 ±1 | 12 ±10 | 78 ±35 | 119 ±44 | 140 ±91 |
|  | Nut assembly | 138 ±56 | 2 ±1 | 10 ±7 | 12 ±7 | 90 ±61 | 4 ±3 | 4 ±3 | 14 ±11 | 30 ±36 | 57 ±51 | 106 ±30 |
|  | Two-arm lifting | 377 ±37 | 25 ±7 | 12 ±6 | 30 ±13 | 12 ±11 | 24 ±10 | 21 ±10 | 19 ±16 | 26 ±13 | 24 ±15 | 33 ±20 |
|  | Peg-in-hole | 520 ±47 | 164 ±63 | 130 ±81 | 154 ±34 | 296 ±90 | 155 ±73 | 154 ±72 | 157 ±55 | 223 ±62 | 215 ±88 | 264 ±81 |

task under Easy, Hard, and Extreme environment shifts respectively. On the task of Nut assembly, the performance boosts are +443%, +133%, and +114%. Similar performance boosts can also be observed on Two-arm lifting and Peg-in-hole tasks as well as between SVEA_C and SVEA_C+SRM models.

# D   Discussions and Relations

**The algorithms of generalization in image-based Reinforcement Learning** CURL [10] utilizes contrastive learning to extract high-level features from image observations. In DrQ [18], random shift are used to regular the Q-function. PAD [6] is trained with a standard RL objective and a self-supervised objective but tested only with the latter. In RAD [11], the use of data augmentation for RL is studied and random crop is found the most effective. In SODA [8], augmentation is decoupled from policy learning. In SVEA [7], only unaugmented observations are used to compute Q-target. In DrAC [13], Raileanu *et al.* attempt to automatically find an effective augmentation for RL task. In SECANT [3], Fan *et al.* point that weak augmentations can improve RL optimization on current environment but fail to provide generalization ability, and strong augmentations is easy to make training divergence. In VAI [15], Wang *et al.* attempt to provide a clear image for testing by extract foregrounds with unsupervised keypoint detection. All data augmentations used in above algorithms are spatial-based, while our method is spectral-based. The proposed method is an augmented algorithm, thus it can be effectively combined with the above naive algorithms (DrQ [18] and RAD [11]) or well-designed algorithms (DrAC [13], SODA [8], SVEA [7] and SECANT [3]) for using data augmentation in RL. Although VAI [15] is the most robust because the background is clear, it is also the most troublesome because additional training of a detection network is required.

**The related works of data augmentations** The examples of spatial-based and spectral-based data augmentations are shown in Figure 7. Random convolution [12] uses a randomly initialized convolutional layer. Random overlay [8] linearly interpolates an observation with an extra image. Random flip [11] flips the image horizontally. Random rotation [4] rotates the image according to a random angle. Cutout [1] randomly erases a patch of the input image. Instead of directly removing a patch, Cut-Noise and Cutmix use random noise and a patch of another image to replace the corresponding input patch, respectively. Compared with random flip and random rotation, the proposed spectral-based data augmentation SRM is invariant to RL tasks. That is, the collected actions and rewards in previous episodes are still available to augmented observation by SRM. As shown in Fig. 7, masking from spatial view (cutout, cut-noise and cutmix) has an obvious downside: the task-related part is easily to be removed or occluded. In contrast, masking from spectral aspect is

able to remain the integrity of observation. For random convolution and random overlay, the proposed SRM could be seamlessly combined with them to further improve the diversity of observations.

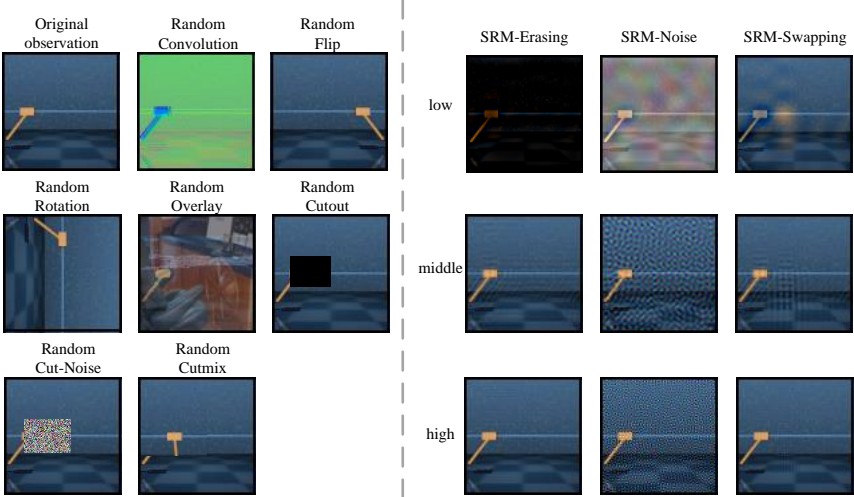

Figure 7: Spatial-based and spectral-based data augmentations.

**The relationship between recently proposed masked image/frequency modeling methods** We compare SRM with several recent works that utilize masked modeling. MAE [9] and SimMIM [17] are concurrent works that share similar idea of reconstructing spatially masked patches on input images. MFM [16] is the frequency domain counterpart of MAE and SimMIM, which masks a portion of frequency components and later learns to reconstruct them. MLR [19] adopts the 3D cube masking on observations and learns to reconstruct latent states under a BYOL-like self-supervised framework. MWM [14] utilizes the spatial masking strategy like MAE and SimMIM, however on convolutional features, to capture fine-grained details within patches. MLR and MWM can be treated as unique feature domain spatial masked modeling methods specially designed for RL tasks. The above five works share the same following spirit (also quoted by [17]):

"What I cannot create, I do not understand." — Richard Feynman

Different from these works, SRM dynamically erases a portion of frequency components and forces the model to do the right prediction without them. Taking again Feynman's wisdom, SRM has a different spirit as follows:

"The first principle is that you must not fool yourself and you are the easiest person to fool."— Richard Feynman

Deep models tend to bias toward certain frequency bands. When frequency distribution changes, they are easily fooled. The key insight of SRM is thus to erase dynamically a wide range of frequency bands so that the model cannot fool itself by only looking at some particular frequency.

## E  Limitation and Future Direction

We have discussed limitation or SRM in the main manuscript that it can not be used in non-image RL tasks. Here we further discuss the case where images are used as inputs. Specifically, one potential limitation of SRM is that it cannot be applied to tasks in which a particular frequency band plays a vital role, to the extent that any modification will cause a failure. For example, in the task of surface defect detection of mechanical parts, low-frequency information reflects the smoothness of a surface, thus any modification will confuse the decision process of an agent. In this case, it is better to select appropriate augmentation techniques carefully or not apply any augmentation at all. However, such a situation is rare compared with other tasks where SRM can benefit.

We also introduce here a promising future research direction on RL policy generalization. The direction comes from an observation that most of RL algorithms including SRM are not robust to

Table 7: Performance under different camera views. The camera azimuth of Drawerworld is adjusted with various angles.

| Method | DrQ | | | DrQ+SRM | | |
|---|---|---|---|---|---|---|
| Texture\Camera azimuth | 180 | 190 | 300 | 180 | 190 | 300 |
| Grid | 82±3 | 73±15 | 0 | 89±4 | 76±10 | 0 |
| Black | 75±7 | 63±9 | 0 | 91±5 | 69±4 | 0 |
| Fabric | 25±4 | 10±6 | 0 | 56±5 | 31±9 | 0 |
| Metal | 79±5 | 71±12 | 0 | 92±2 | 71±7 | 0 |
| Wood | 35±7 | 14±8 | 0 | 72±5 | 29±9 | 0 |

camera view changes. We conducted experiments on DrawerWorld by adjusting camera azimuth to test SRM toward different camera views. The results are in Table 7 (we use azimuth=180 for training). It can be found that DrQ+SRM performs better than DrQ under slightly changed azimuth, yet both models fail to generalize toward large azimuth change. We have also tested on DMControl, where both DrQ+SRM and DrQ drop to near 0 rewards after changing camera ID. The results show that adapting to camera view changes is a much more challenging task than adapting to visual environment shifts. Therefore, future works on viewpoint-robust RL policy generalization will be an interesting direction to explore.

## F   Potential Negative Societal Impacts

Deep reinforcement learning may bring potential security risks. First, due to the uninterpretability of deep neural networks, it is easy to be attacked by artificial perturbations. Second, even highly secure agents strictly restricted in all directions can become dysfunctional when an adversary interferes with perceptual inputs or modifies their rewards. These attributes make agents vulnerable to exploitation by criminals. What's more, when the agent of deep reinforcement learning is applied to healthcare and transportation, people's lives may be at risk once attacked.