# OpenReview forum: "Spectrum Random Masking for Generalization in Image-based Reinforcement Learning"
_NeurIPS.cc/2022/Conference — NeurIPS 2022 Accept_

### Official Review · Reviewer_JBru · 2022-07-10

**Rating:** 7
**Confidence:** 3
**Soundness:** 3 good
**Presentation:** 3 good
**Contribution:** 3 good

**Summary:**

This work presents a new perspective of data augmentation for policy generalization in image-based RL. As different types of spatial corruptions influence the model robustness toward different frequency ranges, the authors conjecture that the observation’s the observation’
Fourier statistics for the same RL task also vary across different environment distributions. Thus, the authors propose to transform an original observation to the frequency domain, augment a portion of the frequency and inverse-transform the augmented frequency map to the pixel domain for agent training. Through such "frequency augmentation"  is shown to be effective in many generalization environments.

**Questions:**

Questions:
1. Will SRM be robust to camera view?
2. What if use the commonly-used encoder of three convolutional layers instead of the very deep 11-conv-layers encoder used in the paper? I expect to see the experiment.

Suggestions:
1. Figure 5 suggests that Erasing is very effective. In other words, directly masking on the frequency map is very effective. It would be better if the authors discuss the relationship between recently popular masked image/frequency modeling [1-5] and the proposed SRM.
2. Better study other mask form such as fan-shaped mask and frequency-band mask, instead of only ring-shaped mask.

[1] He, Kaiming, et al. "Masked autoencoders are scalable vision learners." Proceedings of the IEEE/CVF Conference on Computer Vision and Pattern Recognition. 2022.

[2] Xie, Zhenda, et al. "Simmim: A simple framework for masked image modeling." Proceedings of the IEEE/CVF Conference on Computer Vision and Pattern Recognition. 2022.

[3] Xie, Jiahao, et al. "Masked Frequency Modeling for Self-Supervised Visual Pre-Training." arXiv preprint arXiv:2206.07706 (2022).

[4] Yu T, Zhang Z, Lan C, et al. Mask-based Latent Reconstruction for Reinforcement Learning[J]. arXiv preprint arXiv:2201.12096, 2022.

[5] Seo Y, Hafner D, Liu H, et al. Masked World Models for Visual Control[J]. arXiv preprint arXiv:2206.14244, 2022.

**Limitations:**

The authors have claimed some limitations.

**Strengths And Weaknesses:**

Strengths:
- The motivation is clear and the idea of "frequency augmentation" for RL generalization is novel.
- The experiments and ablation studies are sufficient to show the effectiveness of the proposed method.

Weaknesses:
- Related Work is not very sufficient/detailed.

---

> ### Author Response · Authors · 2022-08-02
> **Response to Reviewer JBru**
>
> Thank you for your motivating review and concrete suggestions. We are encouraged by the positive comments on the motivation and idea of SRM. Detailed responses to your questions are listed as follows.
>
> &nbsp;
>
> > Will SRM be robust to camera view?
>
> Thanks for the interesting question. We conducted experiments on DrawerWorld by adjusting camera azimuth to test SRM toward different camera views. The results are as follows (we use azimuth=180 for training):
>
> |Method|DrQ|||DrQ_SRM|||
> |---|---|---|---|---|---|---|
> |Texture\Camera azimuth|180|190|300|180|190|300|
> |Grid|82±3|73±15|0|89±4|76±10|0|
> |Black|75±7|63±9|0|91±5|69±4|0|
> |Fabric|25±4|10±6|0|56±5|31±9|0|
> |Metal|79±5|71±12|0|92±2|71±7|0|
> |Wood|35±7|14±8|0|72±5|29±9|0|
>
>
> It can be found that DrQ+SRM performs better than DrQ under slightly changed azimuth, yet both models fail to generalize toward large azimuth change.  We have also tested on DMControl, where both DrQ+SRM and DrQ drop to near 0 rewards after changing camera ID. The results show that adapting to camera view changes is a much more challenging task than adapting to visual environment shifts.
>
>
>
> > What if use the commonly-used encoder of three convolutional layers instead of the very deep 11-conv-layers encoder?
>
> Following your advice, we tested the three-layer encoder on DMControl. The results are as follows:
>
> |#Layers||3||||11|||
> |---|---|---|---|---|---|---|---|---|
> |Task||walker_walk||cartpole_swingup||walker_walk||cartpole_swingup|
> |Color_easy|DrQ|803±36||814±22||826±10||852±15|
> ||DrQ_SRM|876±15||840±12||912±21||878±1|
> |Color_hard|DrQ|493±59||564±34||520±91||586±52|
> ||DrQ_SRM|737±45||784±39||806±88||802±28|
> |Video_easy|DrQ|519±60||460±95||682±89||485±105|
> ||DrQ_SRM|765±37||696±48||823±32||740±24|
> |Video_hard|DrQ|94±13||137±23||104±22||138±9|
> ||DrQ_SRM|153±8||197±64||225±29||254±58|
>
> We see that SRM works on both encoders, and the 11-conv-layers encoder always outperforms the 3-conv-layer one due to increased network capacity.
>
> &nbsp;
>
> >It would be better to discuss the relationship between recently popular masked image/frequency modeling [1-5] and the proposed SRM.
>
> Thanks for the valuable suggestion. These works are indeed related to ours and we will add discussions about them in the revised paper. MAE [1] and SimMIM [2] are concurrent works that share similar idea of reconstructing spatially masked patches on input images. MFM [3] is the frequency domain counterpart of MAE and SimMIM, which masks a portion of frequency components and later learns to reconstruct them. MLR[4] adopts the 3D cube masking on observations and learns to reconstruct latent states under a BYOL-like self-supervised framework. MWM [5] utilizes the spatial masking strategy like MAE and SimMIM, however on convolutional features, to capture fine-grained details within patches. MLR and MWM can be treated as unique feature domain spatial masked modeling methods specially designed for RL tasks.
>
> The above five works share the same following spirit (also quoted by [2]):
>
> > **“What I cannot create, I do not understand.” — Richard Feynman**
>
> Different from these works, SRM dynamically erases a portion of frequency components and forces the model to do the right prediction without them. Taking again Feynman's wisdom, SRM has a different spirit as follows:
> > **“The first principle is that you must not fool yourself and you are the easiest person to fool.”— Richard Feynman**
>
> Deep models tend to bias toward certain frequency bands. When frequency distribution changes, they are easily fooled. The key insight of SRM is thus to erase dynamically a wide range of frequency bands so that the model cannot fool itself by only looking at some particular frequency.
>
> &nbsp;
>
> > Better study other mask form such as fan-shaped mask and frequency-band mask.
>
> Thanks for the suggestion. We have done experiments on DMControl walker walk task using fan-shaped masks with different angles, high-pass filters, low-pass filters, and band-pass filters. We use DrQ as the base algorithm. The results are as follows:
>
> |Task\Mask|Drq|SRM|FAN_45|FAN_90|FAN_135|FAN_180|BandPass|LowPass|HighPass|
> |---|---|---|---|---|---|---|---|---|---|
> |color_easy|826±10|912±21|634±24|710±17|645±23|775±32|562±27|818±34|150±20|
> |color_hard|520±91|806±88|384±74|559±71|491±87|588±94|463±62|532±68|136±59|
> |video_easy|682±89|823±32|685±94|586±55|657±79|785±68|502±77|674±87|113±63|
> |video_hard|104±22|225±29|119±16|121±25|185±17|225±20|99±14|62±19|56±17|
>
> We see that fan-shaped masks generally perform worse than SRM. The reason might be that masking a fan-shaped section drops too much information and cannot eliminate any frequency band entirely at the same time. The performance of increases with larger fan angles, indicating that it is better to eliminate the entire frequency band ring. The three band masks also perform worse than SRM, indicating that it is better to mask a wide range of bands instead of leaving only a certain band. We will elaborate these in the revised paper.

---

> > ### Comment · Reviewer_JBru · 2022-08-05
> > **Response to author response**
> >
> > Thank you for the detailed response. I appreciate the experiments and encourage the authors to inlcude these dicussion/analysis in the final revision. It seems that generalization regarding camera view is still a hard problem in policy generalization. Future work on this must be interesting. Besides, the quotes are proper and I appreciate them.

---

> > > ### Author Response · Authors · 2022-08-06
> > > **Thank you for your feedback**
> > >
> > > We sincerely appreciate your positive feedback. We have added extra discussion and analysis in our revised Appendix. The new Appendix now includes 1) more experiments in different environments, 2) discussion on the effectiveness of various alternative spectral mask forms,
> > >  3) detailed comparison with methods [1-5], and 4) indication of future research directions (viewpoint robust generalization).
> > >
> > > We couldn't have gone this far without your detailed response, thank you again.

---

### Official Review · Reviewer_m2cb · 2022-07-10

**Rating:** 6
**Confidence:** 4
**Soundness:** 2 fair
**Presentation:** 3 good
**Contribution:** 2 fair

**Summary:**

This paper focuses on zero-shot policy generalization to new environments in vision-based deep reinforcement learning. It proposes a new data augmentation method by random masking in the frequency domain, dubbed "Spectrum Random Masking (SRM)". SRM can be plugged into existing policy generalization algorithms like DrQ and SVEA. Experimental results are demonstrated on the Distracting DMControl benchmark, where the environment background is replaced by random color or distracting videos.

**Questions:**

As stated in the "Weakness" section above, how does the proposed method work on environments that are more challenging and realistic than the toy Distracting DMControl? For example, autonomous driving, robotics, indoor navigation, or other practically meaningful domains?

**Limitations:**

The limitation that the authors listed is that the method is only applicable to image-based RL tasks and cannot incorporate state information or heavily occluded image observations.

However, I believe that another significant limitation is the applicability of the method to meaningful tasks in the real world, rather than the artificially constructed Distracting DMControl suite. The experimental evidence is insufficient as of the current writing.

**Strengths And Weaknesses:**

# Strengths

1. This paper is quite easy to follow. Fig. 2 and Fig. 3 are very good visualizations to show the intuition of frequency-domain augmentation for policy learning.
2. The proposed Spectrum Rnadom Masking method is conceptually simple and easy to understand and adopt. It is also a plug and play data augmentation module that can be integrated into many policy learning methods. The paper demonstrates experiments on DrQ+SRM and SVEA+SRM.


# Weaknesses

My primary concern with the paper is the experiment section. The only benchmark demonstrated in the paper is Distracting DMControl, which is a set of toy environments with artificial distractions, such as random color and noisy video background. The tasks (such as Half-Cheetah, Walker-run, etc.) are not grounded in any useful real-life scenarios, and the visual distractions are not from naturally occurring sources. Furthermore, Distracting DMControl is already quite saturated, as the baselines methods can achieve close to 100% score in a significant number of tasks.

SECANT [1] includes a number of visual policy generalization benchmarks that feature test-time visual appearance shifts that are representative of real-world applications, such as autonomous driving (CARLA [2]), robotics (Robosuite [3]), and indoor navigation in novel rooms (iGibson [4]). Please consider using these environments that are more realistic and challenging, or other benchmarks of similar calibre.

> Reference
* [1]  SECANT: Self-Expert Cloning for Zero-Shot Generalization of Visual Policies. ICML 2021.
* [2] CARLA: An Open Urban Driving Simulator. 2017.
* [3] robosuite: A Modular Simulation Framework and Benchmark for Robot Learning. 2020.
* [4] IGibson 1.0: a Simulation Environment for Interactive Tasks in Large Realistic Scenes. 2021.

---

> ### Author Response · Authors · 2022-08-02
> **Response to Reviewer m2cb**
>
> Thank you very much for your constructive comments on the paper. We address your main concerns as follows.
>
> > My primary concern with the paper is the experiment section. The only benchmark demonstrated in the paper is Distracting DMControl, which is a set of toy environments with artificial distractions, such as random color and noisy video background.
>
> > How does the proposed method work on environments that are more challenging and realistic than the toy Distracting DMControl? For example, autonomous driving, robotics, indoor navigation, or other practically meaningful domains?
>
> > I believe that another significant limitation is the applicability of the method to meaningful tasks in the real world, rather than the artificially constructed Distracting DMControl suite. The experimental evidence is insufficient as of the current writing.
>
> > SECANT [1] includes a number of visual policy generalization benchmarks that feature test-time visual appearance shifts that are representative of real-world applications. Please consider using these environments that are more realistic and challenging, or other benchmarks of similar calibre.
>
> Thank you for pointing out that the effectiveness of our method should be validated in more challenging and realistic environments. Following your valuable advice, we have conducted more experiments on other environments. Two of our new test environments are drawn from SECANT [1] (**Robosuite [2]**  and **CARLA [3]** ). The last one is **DrawerWorld [4]** which is a robot control environment recommended by another reviewer.
>
> Please refer to our general response posted above for detailed experiment results. The results show that SRM also works effectively in these challenging real-world environment settings.
>
> Specifically, on Robosuite (please refer to the second table in our general response for all reviewers),
> adding our proposed SRM on DrQ (denoted as DrQ+SRM) can boost the performance of DrQ by **+169%**, **+560%**, **+550%** on the DoorOpening task under Easy, Hard, and Extreme environment shifts respectively. On the task of Nut assembly, the performance boosts are +**443%**, **+133%**, and **+114%**. Similar performance boosts can also be observed between SVEA_C and SVEA_C+SRM.
>
> On DrawerWorld [4] (see the first table in our general response), we boost the performance of DrQ by **+35.13%**. Without bells and whistles and special designs, we achieve second best on DrawerWorld (The best model is VAI [4], which uses a complex training pipeline to remove background distractions completely).
>
> On CARLA [3] (the last table in our general response), using SRM can increase the driving distance of an autonomous driving car by **+27.4%** and **+11.9%** with DrQ and SVEA-C as baseline methods, respectively.
>
> We hope these empirical results are sufficient to address your concern. We are also running experiments on more tasks, and we will post the results later in the discussion if they become available.
>
>
> **Reference**
>
> [1] SECANT: Self-Expert Cloning for Zero-Shot Generalization of Visual Policies. ICML 2021.
>
> [2] robosuite: A Modular Simulation Framework and Benchmark for Robot Learning. arXiv:2009.12293, 2020.
>
> [3] CARLA: An Open Urban Driving Simulator. 2017.
>
> [4] Unsupervised Visual Attention and Invariance for Reinforcement Learning. CVPR 2021.

---

> > ### Comment · Reviewer_m2cb · 2022-08-09
> > **Thanks for the additional experiments!**
> >
> > Thanks for the additional experiments, I really appreciate the effort! Please include these results and discussions in your final draft. I have adjusted my score to weak accept accordingly.

---

> > > ### Author Response · Authors · 2022-08-09
> > > **Thank you very much!**
> > >
> > > We sincerely appreciate your advice and recommendations of different environments. Thank you for your effort and supportive feedback. We learned a lot from the review. We have added the discussion in the revised manuscript and will further elaborate on it in the final draft.

---

### Official Review · Reviewer_UTXd · 2022-07-11

**Rating:** 5
**Confidence:** 5
**Soundness:** 3 good
**Presentation:** 3 good
**Contribution:** 2 fair

**Summary:**

This paper proposes a new data augmentation technique that masks out components in the frequency space: the Spectrum Random Masking (SRM). To adapt with SRM, the authors propose new techniques to stabilize the Q value after augmentation. The proposed method achieves higher performance than SVEA on video background generalization tasks.

**Questions:**

1. Do you have a in-depth answer about when the SRM technique should be applied? If I am an RL practitioner, my application wouldn't distinguish whether it is color or video background.
2. Can you test your method on one more task other than DMC-GB? For example, you can use DrawerWorld.

**Limitations:**

The authors mention that when the input is not full image, the proposed method would fail. However, the authors did not mention the limitation even when the inputs are images. But we appreciate the authors for at least mentioning that.

**Strengths And Weaknesses:**

Strengths:
1. The paper is clear and easy to understand.
2. The proposed method is straightforward and easy to implement.
3. The performance is better than SVEA on video background environments.
4. This explores new possiblities about using strong and useful augmentations for RL.

Weaknesses:
1. The proposed method is applicable only when the observation has a wide spectrum. For example, in the color background, the proposed method cannot outperform previous method. This largely compromise the generality of the augmentation and its potential to be applied on real-world tasks.
2. This paper only test on DMC-GB. However, this benchmark is far from enough to demonstrate the generalization ability. For example, all the videos are copy-pasted to the background.

---

> ### Author Response · Authors · 2022-08-02
> **Response to Reviewer UTXd**
>
> Thank you for your comments and suggestions. We think there might be a misunderstanding on the performance of our proposed SRM in the color background. We will re-clarify these concerns with detailed responses in the following.
>
> >The proposed method is applicable only when the observation has a wide spectrum. For example, in the color background, the proposed method cannot outperform previous method.
>
> We are quite confused at first by this comment since our reported results show **exactly the opposite.** In particular, Table.1 in the paper shows that SRM can boost the performance of DrQ, SVEA-C, and SVEA-O under both Color Easy and Color Hard scenarios.
>
> After seeing the comment **The performance is better than SVEA on video background environments,** we figure out there might be a misunderstanding. Specifically, in Table.1 we aim to compare the performance of three baseline RL algorithms (DrQ, SVEA-C, and SVEA-O) **with and without applying our proposed SRM**, therefore we compare DrQ vs. DrQ+SRM,  SVEA-C vs. SVEA-C+SRM, and SVEA-O vs. SVEA-O+SRM. On color backgrounds, SVEA-C+SRM performs the best, which is reasonable since SVEA-C adopts random convolution as additional augmentation. **This performance is still achieved with our SRM.** Our intention is not to cross-compare between DrQ, SVEA-C, and SVEA-O (since they are all baseline methods), but rather to prove that SRM is effective on any of these algorithms. The performance difference between SVEA-C and SVEA-O on color and video backgrounds shows that different RL algorithms have advantages on different kinds of data, yet our **SRM can boost all of these algorithms on both color and video backgrounds.** We are sorry for the possible confusion and we will modify Table.1 to demonstrate the results more clearly.
>
> > Can you test your method on one more task other than DMC-GB? For example, you can use DrawerWorld.
>
> Thanks for your suggestion. We have tested SRM on DrawerWorld [1] and two other new environments (Robosuite [2] and CARLA [3]). Please refer to our general response posted above for detailed results. Experiment results show that SRM is robust and can work effectively on these new challenging environments. We have also cited DrawerWorld [1] and will rearrange the experiment section to include new results in our revised paper.
>
> >Do you have a in-depth answer about when the SRM technique should be applied?
>
> As we analyzed in the paper (L.39-L.51), diverse spectrum patterns exist across environment distributions. Therefore, when facing unknown environments, we must arm the model with the ability to utilize the information lying in all frequency ranges. This is our key motivation for designing SRM, where we erase different frequency ranges dynamically to force the model to learn from all ranges. With SRM, the trained agent does not need to distinguish whether it is color or video background. It can succeed in both backgrounds since SRM leverages the model's ability towards all frequency distributions. **In a word, we can safely apply SRM during training under almost all circumstances.** As shown in Table.1, the performance with SRM is always expected to be better than not using SRM.
>
> Perhaps one necessary decision for us is to choose the base RL algorithm. For example, SVEA-O+SRM performs better on video backgrounds while SVEA-C+SRM works well on color backgrounds. To decide, we need both evaluations of validation data and prior knowledge of the upcoming environment distribution. For instance, if we know the color change is the major distribution shift we will face, we need to choose SVEA-C+SRM instead of SVEA-O+SRM. However, **we could always apply SRM** no matter which algorithm we finally decide on.
>
> >The authors mention that when the input is not a full image, the proposed method would fail. However, the authors did not mention the limitation even when the inputs are images. But we appreciate the authors for at least mentioning that.
>
> Thanks for the suggestion. One potential limitation of SRM is that it cannot be applied to tasks in which a particular frequency band plays a vital role, to the extent that any modification will cause a failure. For example, in the task of surface defect detection of mechanical parts, low-frequency information reflects the smoothness of a surface, thus any modification will confuse the decision process of an agent. In this case, it is better to select appropriate augmentation techniques carefully or not apply any augmentation at all. However, such a situation is rare compared with other tasks where SRM can benefit. We will add this in our revised manuscript and elaborate on it in the appendix.
>
> **References**
>
> [1] Unsupervised Visual Attention and Invariance for Reinforcement Learning. CVPR 2021.
>
> [2] robosuite: A Modular Simulation Framework and Benchmark for Robot Learning. arXiv:2009.12293, 2020.
>
> [3] CARLA: An Open Urban Driving Simulator. 2017.

---

> > ### Comment · Reviewer_UTXd · 2022-08-08
> > **Response to rebuttal**
> >
> > The authors addressed my concerns. While there are still minor issues, I will increase my rating to 5.

---

> > > ### Author Response · Authors · 2022-08-09
> > > **Thank you for your positive response**
> > >
> > > We are grateful for your positive support. We are also very glad to see that we have addressed your concerns. Your recommendation of the new testing environment and the advice of discussing potential limitations are invaluable for us to improve the quality of the manuscript. Please let us know if you have further advice and suggestions. Thank you so much.

---

### Official Review · Reviewer_T6K6 · 2022-07-12

**Rating:** 7
**Confidence:** 4
**Soundness:** 4 excellent
**Presentation:** 4 excellent
**Contribution:** 3 good

**Summary:**

This paper proposes a novel way of performing data augmentation in image-based reinforcement learning by perturbing regions in the frequency domain of the observed images. The method improves upon state-of-the-art data augmentation methods both during training and in generalization.

**Questions:**

- Simple data augmentation methods like cropping and flipping add almost no computational overhead in the RL training loop. How demanding is SRM in computation power?

**Limitations:**

- Typo in Figure 4: average performance is the last column.

**Strengths And Weaknesses:**

Strengths:
- The explanation and visualization of the Spectrum Random Masking method are excellent.
- The proposed approach is a novel and effective addition to the existing data augmentation methods.
- Ablation studies justify the specific implementation choices in the default SRM.
- The experiments are comprehensive.

Weaknesses:
- I would love to see a real-world experiment where data augmentation plays a more important role.

---

> ### Author Response · Authors · 2022-08-02
> **Response to Reviewer T6K6**
>
> We appreciate your efforts for reviewing this paper. We are glad you enjoyed the visualization and analysis of SRM. Please find our detailed response to your questions below:
>
> > I would love to see a real-world experiment where data augmentation plays a more important role.
>
> Thanks for your suggestion.  We have conducted more experiments on additional environments including DrawerWorld, Robosuite (both for robot control), and CARLA (autonomous driving). Please refer to our general response posted above for detailed results. Experiment results show that SRM also contributes positively in these real-world settings and it can improve the existing models by a large margin.
>
>
> >Simple data augmentation methods like cropping and flipping add almost no computational overhead in the RL training loop. How demanding is SRM in computation power?
>
> Thanks for the interesting question. The computation of SRM consists of the following steps:
> 1. Performing 2D Fast Fourier transform (FFT) on input images. FFT has a time complexity of $O(NlogN)$, where $N$ is the number of pixels.
> 2. Generating a 2D binary mask $M$ and masking frequency components calculated from step1 with $M$. This operation has a time complexity of $O(N)$.
> 3. Performing inverse Fourier transform (IFFT) on the masked frequency components to get the augmented output image. IFFT has a time complexity of $O(NlogN)$.
>
> Overall, SRM has a time complexity of $O(NlogN)$, which is higher than cropping and flipping ( both have $O(N)$ complexity, here we assume the crop size is proportional to the image size). However, the time complexity of SRM is still moderate compared with advanced augmentation techniques (such as random convolution).
>
> Experimentally, we test the GPU running time of random cropping (we set the crop size to $64\times 64$), flipping, random convolution, and SRM on a batch of $64\times 3\times 256\times 256$ image data by performing each augmentation $1,000$ times on the same  RTX 3090 GPU. The average time per operation are reported as follows:
> - cropping: 0.5231 ms
> - flipping: 0.1253 ms
> - random convolution: 35.7456 ms
> - SRM: 8.8934 ms
>
> The experiment results also demonstrates that SRM has higher time complexity than cropping and flipping, while being more efficient than random convolution. Overall, the computation time of SRM is less than 10 ms per operation, which is negligible compared to the inference time of the RL agent.
>
> >Typo in Figure 4: average performance is the last column.
>
> Thanks for pointing it out. We have fixed this in the revised manuscript.

---

### Author Response · Authors · 2022-08-02
**To All Reviewers (Additional Experiments on Multiple Environments)**

We thank the reviewers for their time and insightful feedback. In this general response, we would like to address the concerns about the effectiveness of our SRM in more challenging and realistic environments. Specifically, we conduct experiments using 3 additional environments (**DrawerWorld**, **Robosuite**, and **CARLA**) as suggested by the reviewers. DrawerWorld [1] is a realistic texture benchmark for robot manipulation. Robosuite [2] is a robitic simulator with more distracting textures of the table, floor, and objects. CARLA [4] is a driving simulator including various weathers. These additional experimental results further demonstrate the effectiveness of our SRM. Our code will also be submitted as the revised supplementary. The detailed results are as follows:

**Results on DrawerWorld**

The Success Rate (%) of different methods are list below, and the results of SAC, PAD, and VAI are from the VAI paper [1].

|Texture\Method|SAC|PAD|VAI|DrQ(our baseline)|DrQ-SRM(ours)|
|----------------|------|------|-------|------|---------|
|Grid|98±2|84±7|100±0|82±3|89±4|
|Black|95±2|95±3|100±1|75±7|91±5|
|Fabric|2±1|20±6|99±1|25±4|56±5|
|Metal|35±7|81±3|98±2|79±5|92±2|
|Wood|18±5|39±9|94±4|35±7|72±5|
|Average|49.6|63.8|98.2|59.2|80.0|


**The success rates of the DrQ baseline is significantly boosted with SRM under different background textures**, which demonstrates its effectiveness. We also achieve the second best performance among all methods (The best model VAI [1] uses sophisticated keypoint detection and foreground restoration networks to remove the background distraction completely.  In contrast, our SRM is a plug-and-play data augmentation method without any additional model modification. Hence, the comparison with VAI is just  listed for reference.)

&nbsp;

**Results on Robosuite**

We evaluate SRM on Robosuite [2] under progressively harder tasks. The rewards of compared methods are reported in the following table. We adopt DrQ and SVEA_C as our baselines and then add SRM on these two baselines. Results of other methods are reported by SECANT [3].

|Hardness|Tasks\Methods|SECANT|SAC|SAC+crop|DR|NetRand|SAC+IDM|PAD|DrQ (our baseline)|DrQ_SRM (ours)|SVEA_C (our baseline)|SVEA_C_SRM (ours)|
|----------|-----------------|-----------|-------|----------|---------|---------|---------|-------|-------|---------|---------|------------|
|Easy|Door opening|782±93|17±12|10±8|177±163|438±157|3±2|2±1|79±52|213±105|325±184|452±112|
||||||||||||||
||Nut assembly|419±63|3±2|6±5|12±7|242±28|13±12|11±10|16±12|87±27|150±31|251±24|
||||||||||||||
|Hard|Door opening|522±131|11±10|11±7|37±31|133±82|2±1|2±1|15±8|99±75|102±43|154±37|
||||||||||||||
||Nut assembly|437±102|6±7|9±8|33±18|181±53|34±28|24±26|21±15|49±14|74±28|167±47|
||||||||||||||
|Extreme|Door opening|309±147|11±10|6±4|52±46|140±107|2±1|2±1|12±10|78±35|119±44|140±91|
||||||||||||||
||Nut assembly|138±56|2±1|10±7|12±7|90±61|4±3|4±3|14±11|30±36|57±51|106±30|

It can be found that **SRM also boosts baseline methods on Robosuite** and achieves better average performance than other methods except for SECANT. SECANT uses expert policy with weak augmentation data to guide student policy with strong augmentation data, which is orthogonal to our SRM. Hence, the comparison with SECANT is just listed for reference. In theory, SRM could be seamlessly integrated into SECANT for further performance gain.  Due to the training code of SECANT is not available, we will validate it in our future work.

&nbsp;

**Results on CARLA**

We follow the same settings as in SECANT[3] on CARLA and report the distance (m) traveled in a town without collision. The results are as follows:

|Weather|SECANT|SAC|SAC+crop|DR|NetRand|SAC+IDM|PAD|DrQ (our baseline)|DrQ-SRM (ours)|SVEA_C (our baseline)|SVEA_C_SRM (ours)|
|-------------------|---------|-----------|----------|---------|---------|---------|---------|---------|---------|---------|------------|
|Clear noon(training)|596±77|282±71|684±114|486±141|648±61|582±96|632±126|586±108|671±149|616±97|675±146|
|Wet sunset|397±99|57±14|26±18|9±11|284±84|25±11|36±12|34±15|93±21|297±121|354±95|
|Wet cloudy noon|629±204|180±45|283±85|595±260|557±107|433±105|515±52|393±89|490±92|519±112|576±108|
|Soft rain sunset|435±66|55±28|38±25|25±41|251±104|36±32|41±37|97±67|160±82|208±103|239±93|
|Mid rain sunset|470±202|50±8|37±16|24±24|233±117|42±23|32±21|76±48|136±78|205±105|240±85|
|Hard rain noon|541±96|237±85|235±129|341±96|458±72|156±194|308±141|289±93|342±84|429±83|461±94|

As similar to Robosuite, the results show that **SRM also improves the baseline models' generalization ability on CARLA**.

&nbsp;

**References**

[1] Unsupervised Visual Attention and Invariance for Reinforcement Learning. CVPR 2021.

[2] robosuite: A Modular Simulation Framework and Benchmark for Robot Learning. arXiv, 2020.

[3] SECANT: Self-Expert Cloning for Zero-Shot Generalization of Visual Policies. ICML 2021.

[4] CARLA: An Open Urban Driving Simulator. 2017.

---

### Author Response · Authors · 2022-08-06
**Revised Manuscript and Appendix**

We have uploaded the revised manuscript and appendix.  Please refer to the blue-colored text for newly added contents.

Specifically, we have fixed some typos in the main manuscript, and the new appendix (in supplementary material) now includes:
+ Experiments in different challenging environments. Please note that we have included more experiment results in the appendix despite the results posted in our response threads.
+ Discussions on the effectiveness of various alternative spectral mask shapes.
+ Detailed comparisons with some most recently proposed masked image modeling methods.
+ Potential limitations of our method when inputs are images.
+ Indication of possible future research directions (viewpoint robust generalization).
+ An anonymous link providing our source code zip file.

We thank all reviewers for their comments and suggestions to help us improve this paper.

---

### Meta-Review · Area_Chair_iDm5 · 2022-08-25

**Recommendation:** Accept
**Confidence:** Certain

**Metareview:**

The authors have introduced a method of data augmentation for image-based reinforcement learning that performs masking in the frequency domain, combined with techniques for stabilizing Q-learning, to achieve improved performance on a number of DMControl Generalization Benchmark tasks.

There was agreement among the reviewers that this work is novel and technically sound, and their concerns were mainly related to the breadth of tasks explored in the initial submission. During the review process the authors have gone to considerable effort to introduce new tasks (e.g. DrawerWorld, Robosuite and CARLA) and the improved performance of their method appears to generalize well. I believe that this work will be of broad interest to the RL community and recommend it for acceptance.

**Award:**

No

---

### Decision · Program_Chairs · 2022-09-14

Accept